# QGen: On the Ability to Generalize in Quantization Aware Training

## Abstract

Quantization lowers memory usage, computational requirements, and latency by utilizing fewer bits to represent model weights and activations. In this work, we investigate the generalization properties of quantized neural networks, a characteristic that has received little attention despite its implications on model performance. In particular, first, we develop a theoretical model for quantization in neural networks and demonstrate how quantization functions as a form of regularization. Second, motivated by recent work connecting the sharpness of the loss landscape and generalization, we derive an approximate bound for the generalization of quantized models conditioned on the amount of quantization noise. We then validate our hypothesis by experimenting with over 2000 models trained on CIFAR-10, CIFAR-100, and ImageNet datasets on convolutional and transformer-based models.

## 1 Introduction

The exceptional growth of technology involving deep learning has made it one of the most promising technologies for applications such as computer vision, natural language processing, and speech recognition. The ongoing advances in these models consistently enhance their capabilities, yet their improving performances often come at the price of growing complexity and an increased number of parameters. The increasing complexity of these models poses a challenge to the deployment in production systems due to higher operating costs, greater memory requirements, and longer response times. Quantization Courbariaux et al. (2015); Hubara et al. (2016); Polino et al. (2018); Zhou et al. (2017); Jacob et al. (2018); Krishnamoorthi (2018) is one of the prominent techniques that have been developed to reduce the model sizes and/or reduce latency. Model quantization represents full precision model weights and/or activations using fewer bits, resulting in models with lower memory usage, energy consumption, and faster inference. Quantization has gained significant attention in academia and industry. Especially with the emergence of the transformer Vaswani et al. (2017) model, quantization has become a standard technique to reduce memory and computation requirements.

The impact on accuracy and the benefits of quantization, such as memory footprint and latency, is well studied Courbariaux et al. (2015); Li et al. (2017); Gholami et al. (2021). These studies are mainly driven by the fact that modern hardware is faster and more energy efficient in low-precision (byte, sub-byte) arithmetic compared to the floating point counterpart. Despite its numerous benefits, quantization may adversely impact accuracy.

Hence, substantial research efforts on quantization revolve around addressing the accuracy degradation resulting from lower bit representation. This involves analyzing the model's convergence qualities for various numerical precisions and studying their impacts on gradients and network updates Li et al. (2017); Hou et al. (2019).

In this work, we delve into the generalization properties of quantized neural networks. This key aspect has received limited attention despite its significant implications for the performance of models on unseen data. This factor becomes particularly important for safety-critical Gambardella et al. (2019); Zhang et al. (2022b) applications. While prior research has explored the performance of neural networks in adversarial settings Gorsline et al. (2021); Lin et al. (2019); Galloway et al. (2017) and investigated how altering the number

of quantization bits during inference can affect model performanceChmiel et al. (2020); Bai et al. (2021), there is a lack of systematic studies on the generalization effects of quantization using standard measurement techniques.

This work studies the effects of different quantization levels on model generalization, training accuracy, and training loss. First, in Section 3, we model quantization as a form of noise added to the network weights. Subsequently, we demonstrate that this introduced noise serves as a regularizing agent, with its degree of regularization directly related to the bit precision. Consistent with other regularization methods, our empirical studies further support the claim that each model requires precise tuning of its quantization level, as models achieve optimal generalization at varying quantization levels. On the generalization side, in Section 4, we show that quantization could help the optimization process convergence to minima with lower sharpness when the scale of quantization noise is bounded. This is motivated by recent works of Foret et al. (2021); Keskar et al. (2016), which establish connections between the sharpness of the loss landscape and generalization. We then leverage a variety of recent advances in the field of generalization measurement Jiang et al. (2019); Dziugaite et al. (2020), particularly sharpness-based measures Keskar et al. (2016); Dziugaite & Roy (2017); Neyshabur et al. (2017), to verify our hypothesis for a wide range of vision problems with different setups and model architectures. Finally, in this section, we present visual demonstrations illustrating that models subjected to quantization have a flatter loss landscape.

After establishing that lower-bit-quantization results in improved flatness in the loss landscape, we study the connection between the achieved flatness of the lower-bit-quantized models and generalization. Our method estimates the model's generalization on a given data distribution by measuring the difference between the loss of the model on training data and test data. To do so, we train a pool of almost 2000 models on CIFAR-10 and CIFAR-100 Krizhevsky (2009) and Imagenet-1K Deng et al. (2009) datasets, and report the estimated generalization gap. Furthermore, we conclude our experiments by showing a practical use case of model generalization, in which we evaluate the vision models under severe cases when the input to the model is corrupted. This is achieved by measuring the generalization gap for quantized and full precision models when different types of input noise are used, as introduced in Hendrycks & Dietterich (2019). Our main contributions can be summarized as follows:

- Our theoretical analysis on simplified models suggests that quantization can be seen as a regularizer.

- We empirically show that there exists a quantization level at which the quantized model converges to a flatter minimum than its full-precision model.

- We empirically demonstrate that quantized models show a better generalization gap on distorted data.

## 2 Related Works

### 2.1 Regularization Effects of Quantization

Since the advent of BinaryConnect Courbariaux et al. (2015) and Binarized Neural Networks Hubara et al. (2016), which were the first works on quantization, the machine learning community has been aware of the generalization effects of quantization, and the observed generalization gains have commonly been attributed to the implicit regularization effects that the quantization process may impose. This pattern is also observed in more recent works such as Mishchenko et al. (2019); Xu et al. (2018); Chen et al. (2021). Even though these studies have empirically reported some performance gain as a side-product of quantization, they lack a well-formed analytical study.

Viewing quantization simply as regularization is relatively intuitive, and to the best of our knowledge, the only work so far that has tried to study this behavior formally is the recent work done in Zhang et al. (2022a), where the authors provide an analytical study on how models with stochastic binary quantization can have a smaller generalization gap compared to their full precision counterparts. The authors propose a *quasi-neural network* to approximate the effect of binarization on neural networks. They then derive the neural tangent kernel Jacot et al. (2018); Bach (2017) for the proposed quasi-neural network approximation. With

this formalization, the authors show that binary neural networks have lower capacity, hence lower training accuracy, and a smaller generalization gap than their full precision counterparts. However, this work is limited to the case of simplified binarized networks and does not study the wider quantization space, and their supporting empirical studies are done on MNIST and Fashion MNIST datasets with no studies done on larger scale more realistic problems. Furthermore, the Neural Tangent Kernel (NTK) analysis requires strong assumptions such as an approximately linear behaviour of the model during training which may not hold in practical setups.

## 2.2 Generalization and Complexity Measures

Generalization refers to the ability of machine learning models to perform well on unseen data beyond the training set. Despite the remarkable success and widespread adoption of deep neural networks across various applications, the factors influencing their generalization capabilities and the extent to which they generalize effectively are still unclear Jiang et al. (2019); Recht et al. (2019).

Minimization of the common loss functions (e.g., cross=entropy and its variants) on the training data does not necessarily mean the model would generalize well Foret et al. (2021); Recht et al. (2019), especially since the recent models are heavily over-parameterized and they can easily overfit the training data. InZhang et al. (2021), the authors demonstrate neural networks' vulnerability to poor generalization by showing they can perfectly fit randomly labeled training data. This is due to the complex and non-convex landscape of the training loss. Numerous works have tried to either explicitly or implicitly solve this overfitting issue using optimizer algorithms Kingma & Ba (2014); Martens & Grosse (2015), data augmentation techniques Cubuk et al. (2018), and batch normalization Ioffe & Szegedy (2015), to name a few.

**So the question remains: what is the best indicator of a model's generalization ability?** Proving upper bounds on the test error Neyshabur et al. (2017); Bartlett et al. (2017) has been the most direct way of studying the ability of models to generalize; however, the current bounds are not tight enough to indicate the model's ability to generalize Jiang et al. (2019). Therefore, several recent works have preferred the more empirical approaches of studying generalization Keskar et al. (2016); Liang et al. (2019). These works introduce a complexity measure, a quantity that monotonically relates to some aspect of generalization. Specifically, lower complexity measures correspond to neural networks with improved generalization capacity. Many complexity measures are introduced in the literature, but each of them has typically targeted a limited set of models on toy problems. However, recent work in Jiang et al. (2019) followed by Dziugaite et al. (2020) performed an exhaustive set of experiments on the CIFAR-10 and SVHN Netzer et al. (2011) datasets with different model backbones and hyper-parameters to identify the measures that correlate best with generalization. Both of these large-scale studies show that sharpness-based measures are the most effective. The sharpness-based measures are derived either from measuring the average flatness around a minimum through adding Gaussian perturbations (PAC-Bayesian bounds McAllester (1999); Dziugaite & Roy (2017)) or from measuring the worst-case loss, i.e., sharpness Keskar et al. (2016); Dinh et al. (2017).

The effectiveness of sharpness-based measures has also inspired new training paradigms that penalize the loss of landscape sharpness during training Foret et al. (2021); Du et al. (2022); Izmailov et al. (2018). In particular, Foret et al. (2021) introduced the Sharpness-Aware-Minimization (SAM), which is a scalable and differentiable algorithm that helps models to converge and reduce the model sharpness. It is also worth mentioning here that some recent works Liu et al. (2021); Wang et al. (2022) assume that the discretization and gradient estimation processes, which are common in quantization techniques, might cause loss fluctuations that could result in a sharper loss landscape. Then they couple quantization with SAM and report improved results; however, our findings in Section 4 suggest the opposite. The quantized models in our experiments exhibit improved loss landscape flatness compared to their full precision counterparts.

# 3 Mathematical Model for Quantization

Throughout this paper, we will denote vectors as $\boldsymbol{x}$, the scalars as $x$, and the sets as $\mathcal{X}$. Furthermore, $\perp\!\!\!\perp$ denotes independence. Given a distribution $\mathcal{D}$ for the data space, our training dataset $\mathcal{S}$ is a set of i.i.d. samples drawn from $\mathcal{D}$. The typical ML task tries to learn models $f(.)$ parametrized by weights $\boldsymbol{w}$ that can

minimize the training set loss $\mathcal{L}_{\mathcal{S}}(\boldsymbol{w}) = \frac{1}{|\mathcal{S}|} \sum_{i=1}^{|\mathcal{S}|} l(f(\boldsymbol{w}, \boldsymbol{x}_i), \boldsymbol{y}_i)$ given a loss function $l(.)$ and $(\boldsymbol{x}_i, \boldsymbol{y}_i)$ pairs in the training data.

To quantize our deep neural networks, we utilize Quantization Aware Training (QAT) methods similar to Learned Step-size Quantization (LSQ) Esser et al. (2020) for CNNs and Variation-aware Vision Transformer Quantization (VVTQ) Xijie Huang & Cheng (2023) for ViT models. Specifically, we apply the per-layer quantization approach, in which, for each target quantization layer, we learn a step size $s$ to quantize the layer weights. Therefore, given the weights $\boldsymbol{w}$, scaling factor $s \in \mathbb{R}$ and $b$ bits to quantize, the quantized weight tensor $\hat{\boldsymbol{w}}$ and the quantization noise $\Delta$ can be calculated as below:

$$\bar{\boldsymbol{w}} = \lfloor clip(\frac{\boldsymbol{w}}{s}, -2^{b-1}, 2^{b-1} - 1) \rceil \tag{1}$$

$$\hat{\boldsymbol{w}} = \bar{\boldsymbol{w}} \times s \tag{2}$$

$$\Delta = \boldsymbol{w} - \hat{\boldsymbol{w}}, \tag{3}$$

where the $\lfloor \boldsymbol{z} \rceil$ rounds the input vector $\boldsymbol{z}$ to the nearest integer vector, $clip(r, z_1, z_2)$ function returns $r$ with values below $z_1$ set to $z_1$ and values above $z_2$ set to $z_2$, and $\hat{\boldsymbol{w}}$ shows a quantized representation of the weights at the same scale as $\boldsymbol{w}$.

### 3.1 Theoretical Analysis

For simplicity, let us consider a regression problem where the mean square error loss is defined as,

$$\mathcal{L} = \mathbb{E}_{p(\boldsymbol{x}, \boldsymbol{y})}[\|\hat{\boldsymbol{y}} - \boldsymbol{y}\|_2^2], \tag{4}$$

where $\boldsymbol{y}$ is the target, and $\hat{\boldsymbol{y}} = f(\boldsymbol{x}, \boldsymbol{w})$ is the output of the network $f$ parameterized by $\boldsymbol{w}$.

For uniform quantization, the quantization noise $\Delta$ can be approximated by the uniform distribution $\Delta \sim \mathcal{U}[\frac{-\delta}{2}, \frac{\delta}{2}]$ where $\delta$ is the width of the quantization bin and $\mathcal{U}$ is the uniform distribution Défossez et al. (2021); Widrow et al. (1996); Agustsson & Theis (2020).

Consequently, a quantized neural network effectively has the following loss,

$$\tilde{\mathcal{L}} = \mathbb{E}_{p(\boldsymbol{x}, \boldsymbol{y}, \Delta)}[\|\hat{\boldsymbol{y}^q} - \boldsymbol{y}\|_2^2], \tag{5}$$

where $\hat{\boldsymbol{y}^q} = f(\boldsymbol{x}, \boldsymbol{w} + \Delta)$.

We can apply a first-order Taylor approximation,

$$f(\boldsymbol{x}, \boldsymbol{w} + \Delta) \approx f(\boldsymbol{x}, \boldsymbol{w}) + \Delta^\top \nabla_{\boldsymbol{w}} f(\boldsymbol{x}, \boldsymbol{w}) \tag{6}$$

This does not invalidate our results but rather provides a close approximation. Thus, $\hat{\boldsymbol{y}^q} \approx \hat{\boldsymbol{y}_i} + \Delta^\top \nabla_{\boldsymbol{w}} \hat{\boldsymbol{y}}$. Re-writing, the expectation on $\tilde{\mathcal{L}}$,

$$\begin{aligned}
\tilde{\mathcal{L}} &= \mathbb{E}_{p(\boldsymbol{x}, \boldsymbol{y}, \Delta)}[(\hat{\boldsymbol{y}^q} - \boldsymbol{y})^2] \\
&= \mathbb{E}_{p(\boldsymbol{x}, \boldsymbol{y}, \Delta)}[((\hat{\boldsymbol{y}} + \Delta^\top \nabla_{\boldsymbol{w}} \hat{\boldsymbol{y}}) - \boldsymbol{y})^2] \\
&= \mathbb{E}_{p(\boldsymbol{x}, \boldsymbol{y}, \Delta)}[(\hat{\boldsymbol{y}} - \boldsymbol{y})^2 + \|\Delta^\top \nabla_{\boldsymbol{w}} \hat{\boldsymbol{y}}\|_2^2 + 2(\hat{\boldsymbol{y}} - \boldsymbol{y})(\Delta^\top \nabla_{\boldsymbol{w}} \hat{\boldsymbol{y}})] \\
&= \mathcal{L} + \mathbb{E}_{p(\boldsymbol{x}, \boldsymbol{y}, \Delta)}[\|\Delta^\top \nabla_{\boldsymbol{w}} \hat{\boldsymbol{y}}\|_2^2] + \mathbb{E}_{p(\boldsymbol{x}, \boldsymbol{y}, \Delta)}[2(\hat{\boldsymbol{y}} - \boldsymbol{y})(\Delta^\top \nabla_{\boldsymbol{w}} \hat{\boldsymbol{y}})]
\end{aligned}$$

Since $\Delta \perp\!\!\!\perp \nabla_{\boldsymbol{w}} \hat{\boldsymbol{y}}$, and $\mathbb{E}_{p(\Delta)}[\Delta] = 0$[1], the last term on the right-hand side is zero. Thus we have,

$$\tilde{\mathcal{L}} = \mathcal{L} + \mathbb{E}_{p(\boldsymbol{x}, \boldsymbol{y}, \Delta)}[\|\Delta^\top \nabla_{\boldsymbol{w}} \hat{\boldsymbol{y}}\|_2^2] = \mathcal{L} + \mathcal{R}(\Delta), \tag{7}$$

---

[1]Note that we only require the quantization noise distribution $\Delta$ to have $\mathbb{E}_{p(\Delta)}[\Delta] = 0$. We do not explicitly use the assumption of $\Delta$ coming from a uniform distribution. Thus, for any zero mean noise distribution, the above proof holds.

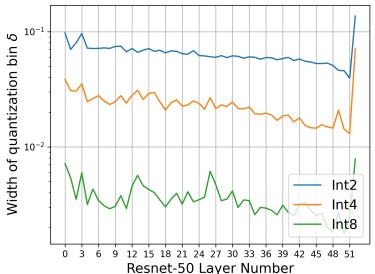 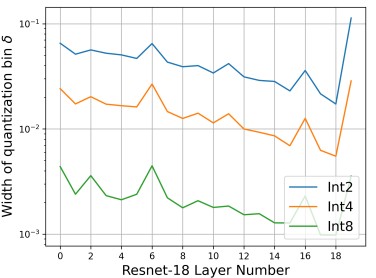 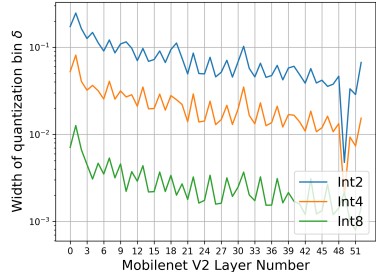

Figure 1: Width of the quantization bin for the weight tensor of each layer in ResNet-50, ResNet-18, and MobileNet V2 trained on the ImageNet dataset. We used different quantization levels and in all cases, the induced quantization noise is significantly higher when a lower bit resolution value is used.

where $\mathcal{R}(\boldsymbol{\Delta})$ can be viewed as a regularization function. This means that minimizing $\tilde{\mathcal{L}}$ is equivalent to minimizing the loss of a non-quantized neural network with gradient norm regularization. Given a quantization method like LSQ or VVTQ, we know that the quantization error ($\boldsymbol{\Delta}$) is a function of the quantization level ($\delta$). As a result, $\mathcal{R}$ is also a function of the quantization level. Thus, the quantization level should be viewed as a hyper-parameter that controls the regularization level. Similar to other regularization methods in deep learning, this hyper-parameter should also be carefully tuned for the best generalization performance.

To study the relation between $\mathcal{R}(\boldsymbol{\Delta})$ and quantization level, we ran some experiments. Figure 1 illustrates the width of the quantization bin per layer for three different architectures trained on ImageNet Deng et al. (2009). As it can be seen, the lower the number of quantization bits, the larger the scale of step size, $s$, is. And as Equations 1 to 3 indicate, $s$ is equivalent to the width of the quantization bin. Hence lower-bit quantization causes quantization bins to be wider as the number of potential representations becomes limited, which results in higher regularization and training losses. In our experiments, this trend was consistent across various vision tasks and model architectures, allowing us to affirm that lower-bit-resolution quantization (with greater $\delta$) results in increased training losses, as shown in Equation 7. This indicates that the level of quantization dictates the degree of regularization introduced to the network. Furthermore, our empirical investigation, encompassing nearly 2000 models trained on the CIFAR-10, CIFAR-100, and ImagenNet-1k datasets, confirms this observation. The findings are detailed in Table 1.

## 4 Analyzing Loss Landscapes in Quantized Models and Implications for Generalization

A low generalization gap is a desirable characteristic of deep neural networks. It is common in practice to estimate the population loss of the data distribution $\mathcal{D}$, i.e. $\mathcal{L}_{\mathcal{D}}(\boldsymbol{w}) = \mathbb{E}_{(\boldsymbol{x},\boldsymbol{y})\sim\mathcal{D}}[l(f(\boldsymbol{w},\boldsymbol{x}),\boldsymbol{y})]$, by utilizing $\mathcal{L}_{\mathcal{S}}(\boldsymbol{w})$ as a proxy, and then minimizing it by gradient descent-based optimizers. However, given that modern neural networks are highly over-parameterized and $\mathcal{L}_{\mathcal{S}}(\boldsymbol{w})$ is commonly non-convex in $\boldsymbol{w}$, the optimization process can converge to local or even global minima that could adversely affect the generalization of the model (i.e. with a significant gap between $\mathcal{L}_{\mathcal{S}}(\boldsymbol{w})$ and $\mathcal{L}_{\mathcal{D}}(\boldsymbol{w})$) Foret et al. (2021).

Motivated by the connection between the sharpness of the loss landscape and generalization Keskar et al. (2016), in Foret et al. (2021) the authors proposed the Sharpness-Aware-Minimization (SAM) technique, in which they propose to learn the weights $\boldsymbol{w}$ that result in a flat minimum with a neighborhood of low training loss values characterized by $\rho$. Especially, inspired by the PAC-Bayesian generalization bounds, they were able to prove that for any $\rho > 0$, with high probability over the training dataset $\mathcal{S}$, the following inequality holds:

$$\mathcal{L}_{\mathcal{D}}(\boldsymbol{w}) \leq \max_{||\boldsymbol{\epsilon}||_2 \leq \rho} \mathcal{L}_{\mathcal{S}}(\boldsymbol{w} + \boldsymbol{\epsilon}) + h(||\boldsymbol{w}||_2^2/\rho^2), \tag{8}$$

Table 1: Generalization gaps of quantized and full-precision models on CIFAR-10, CIFAR-100, and ImagenNet-1k datasets. For NiN models, each cell in the Table represents the mean of that corresponding metric over 243 samples, achieved from training NiN models over variations of 5 common hyperparameters.

| | Model | Precision | Train Acc | Test Acc | Train Loss | Test Loss | Generalization |
|---|---|---|---|---|---|---|---|
| CIFAR-10 | NiN | FP32 | 97.61 | 88.05 | 0.103 | 0.405 | 0.302 |
| | | Int8 | 97.5 | 88.01 | 0.106 | 0.407 | 0.301 |
| | | Int4 | 96.9 | 87.7 | 0.125 | 0.413 | 0.288 |
| | | Int2 | 93.4 | 86.11 | 0.222 | 0.446 | 0.224 |
| CIFAR-100 | NiN | FP32 | 95.28 | 63.48 | 0.207 | 1.687 | 1.48 |
| | | Int8 | 95.17 | 63.44 | 0.211 | 1.685 | 1.469 |
| | | Int4 | 93.5 | 63.19 | 0.271 | 1.648 | 1.38 |
| | | Int2 | 81.21 | 62.11 | 0.676 | 1.537 | 0.859 |
| Imagenet-1K | DeiT-T | FP32 | 73.75 | 71.38 | 1.38 | 2.48 | 1.1 |
| | | Int8 | 76.3 | 75.54 | 0.99 | 1.98 | 0.98 |
| | | Int4 | 74.71 | 72.31 | 1.08 | 2.07 | 0.99 |
| | | Int2 | 59.73 | 55.35 | 1.83 | 2.81 | 0.98 |
| | Swin-T | FP32 | 83.39 | 80.96 | 0.516 | 1.48 | 0.964 |
| | | Int8 | 85.21 | 82.48 | 0.756 | 1.56 | 0.80 |
| | | Int4 | 84.82 | 82.42 | 0.764 | 1.59 | 0.82 |
| | | Int2 | 78.76 | 77.66 | 0.941 | 1.84 | 0.89 |
| | ResNet-18 | FP32 | 69.96 | 71.49 | 1.18 | 2.23 | 1.05 |
| | | Int8 | 73.23 | 73.32 | 1.28 | 2.10 | 0.82 |
| | | Int4 | 71.34 | 71.74 | 1.26 | 2.18 | 0.92 |
| | | Int2 | 67.1 | 68.58 | 1.38 | 2.16 | 0.78 |

where $h : \mathbb{R}_+ \to \mathbb{R}_+$ is a strictly increasing function. Even though the above theorem is for the case where the $L2$-norm of $\boldsymbol{\epsilon}$ is bounded by $\rho$ and the adversarial perturbations are utilized to achieve the worst-case loss, the authors empirically show that in practice, other norms in $[1, \infty]$ and random perturbations for $\boldsymbol{\epsilon}$ can also achieve some levels of flatness; however, they may not be as effective as the $L2$-norm coupled with the adversarial perturbations.

Extending on the empirical studies of Foret et al. (2021), we relax the $L2$-norm condition of Equation 8, and consider the $L_\infty$-norm instead, resulting in:

$$\mathcal{L}_{\mathcal{D}}(\boldsymbol{w}) \leq \max_{||\boldsymbol{\epsilon}||_\infty \leq \rho} \mathcal{L}_{\mathcal{S}}(\boldsymbol{w} + \boldsymbol{\epsilon}) + h(||\boldsymbol{w}||_2^2/\rho^2) \tag{9}$$

Furthermore, given small values of $\rho > 0$, for any noise vector $\boldsymbol{\delta}$ such that $||\boldsymbol{\delta}||_\infty \leq \rho$, the following inequality holds in practice for a local minimum characterized by $\boldsymbol{w}$, as also similarly depicted in Equation 7 where $\boldsymbol{\delta}$ corresponds to the quantization noise, $\boldsymbol{\Delta}$; however, this inequality may not necessarily hold for every $\boldsymbol{w}$:

$$\mathcal{L}_{\mathcal{S}}(\boldsymbol{w}) \leq \mathcal{L}_{\mathcal{S}}(\boldsymbol{w} + \boldsymbol{\delta}) \leq \max_{||\boldsymbol{\epsilon}||_\infty \leq \rho} \mathcal{L}_{\mathcal{S}}(\boldsymbol{w} + \boldsymbol{\epsilon}), \tag{10}$$

For small values of $\rho$ close to 0, and a given $\boldsymbol{w}$ we can approximate,

$$\max_{||\boldsymbol{\epsilon}||_\infty \leq \rho} \mathcal{L}_{\mathcal{S}}(\boldsymbol{w} + \boldsymbol{\epsilon}) \tag{11}$$

in Equation 9 with $\mathcal{L}_{\mathcal{S}}(\boldsymbol{w} + \boldsymbol{\delta})$. As a result, for small positive values of $\rho$, we have:

$$\mathcal{L}_{\mathcal{D}}(\boldsymbol{w}) \leq \mathcal{L}_{\mathcal{S}}(\boldsymbol{w} + \boldsymbol{\delta}) + h(||\boldsymbol{w}||_2^2/\rho^2), \tag{12}$$

and finally, moving the $\mathcal{L}_{\mathcal{S}}(\boldsymbol{w} + \boldsymbol{\delta})$ to the left-hand-side in Equation 12, will give us:

$$\mathcal{L}_{\mathcal{D}}(\boldsymbol{w}) - \mathcal{L}_{\mathbb{S}}(\boldsymbol{w} + \boldsymbol{\delta}) \leq h(||\boldsymbol{w}||_2^2/\rho^2). \tag{13}$$

The above inequality formulates an approximate bound for values of $\rho > 0$ close to 0 on the generalization gap for a model parametrized by $\boldsymbol{w}$; given the nature of function $h(.)$, the higher the value $\rho$ is the tighter the generalization bound becomes.

As shown in Section 3, for quantization techniques with a constant quantization bin width, we have $||\boldsymbol{\Delta}||_\infty \leq \frac{\delta}{2}$, where $\boldsymbol{\Delta}$ is the quantization noise, and $\delta$ is the width of the quantization bin. Replacing the quantization equivalent terms in Equation 13 yields:

$$\mathcal{L}_{\mathcal{D}}(\boldsymbol{w}) - \mathcal{L}_{\mathbb{S}}(\boldsymbol{w} + \boldsymbol{\Delta}) \leq h(4||\boldsymbol{w}||_2^2/\delta^2). \tag{14}$$

We now state the following hypothesis for quantization techniques based on Equation 14:

**Hypothesis 1 (H1)** *Let $\boldsymbol{w}$ be the set of weights in the model, $\boldsymbol{w^q}$ be the set of quantized weights, $\delta$ be the width of quantization bin and $g(.)$ be a function that measures the sharpness of a minima, we have,*

1. *Having a bounded $\boldsymbol{\Delta}$ with $||\boldsymbol{\Delta}||_\infty \leq \frac{\delta}{2}$, there exist a $\boldsymbol{\Delta}$ where, for quantized model parameterized by $w^1$ obtained through QAT and full precision model parameterized by $w^2$ we have: $g(w^1) \leq g(w^2)$*

(1) implies that quantization helps the model converge to flatter minima with lower sharpness. As discussed in Section 3 and illustrated in Figure 1, since lower bit quantization corresponds to higher $\delta$, therefore, lower bit resolution quantization results in better flatness around the minima. However, as described by 7, the $\delta$ is a hyperparameter for the induced regularization. Hence, not all quantization levels will result in flatter minima and improved generalization.

In the rest of this Section, we report the results of our exhaustive set of empirical studies regarding the generalization qualities of quantized models; in Section 4.1, for different datasets and different backbones (models) we study the flatness of the loss landscape of the deep neural networks under different quantization regimens, in Section 4.2 we measure and report the generalization gap of the quantized models for a set of almost 2000 vision models, and finally in Section 4.3 using corrupted datasets, we study the real-world implications that the generalization quality can have and how different levels of quantization perform under such scenarios.

Table 2: $L2$-norm of the network weights; all the weights are first flattened into a $1D$ vector and then the $L2$-norm is measured. The huge difference among the values of different quantization levels indicates that the magnitude of the weights should also be considered when comparing the flatness. Flatness is correlated with the scale of perturbations that the weights can bear without too much model performance degradation. NiN $(d \times w)$ refers to a fully convolutional Network-in-Network Lin et al. (2013) model with a depth multiplier of $d$ and a width multiplier of $w$; the base depth is 25.

| Dataset | Model | Int2 | Int4 | Int8 | FP32 |
|---------|-------|------|------|------|------|
| **CIFAR-10** | NiN (4x10) | 47.263 | 54.291 | 53.804 | 130.686 |
| | NiN (4x12) | 43.039 | 46.523 | 46.750 | 73.042 |
| | ResNet-18 | 44.264 | 48.227 | 47.368 | 59.474 |
| | ResNet-50 | 45.011 | 238.117 | 48.149 | 97.856 |
| **CIFAR-100** | NiN (5x10) | 60.981 | 60.707 | 60.905 | 190.414 |
| | NiN (5x12) | 82.230 | 87.931 | 87.307 | 163.768 |
| | ResNet-18 | 48.120 | 55.027 | 54.735 | 125.164 |
| | ResNet-50 | 75.739 | 82.788 | 79.603 | 148.298 |
| **ImageNet-1K** | ResNet-18 | 78.291 | 84.472 | 85.162 | 415.004 |
| | ResNet-50 | 214.055 | 213.035 | 212.624 | 379.465 |

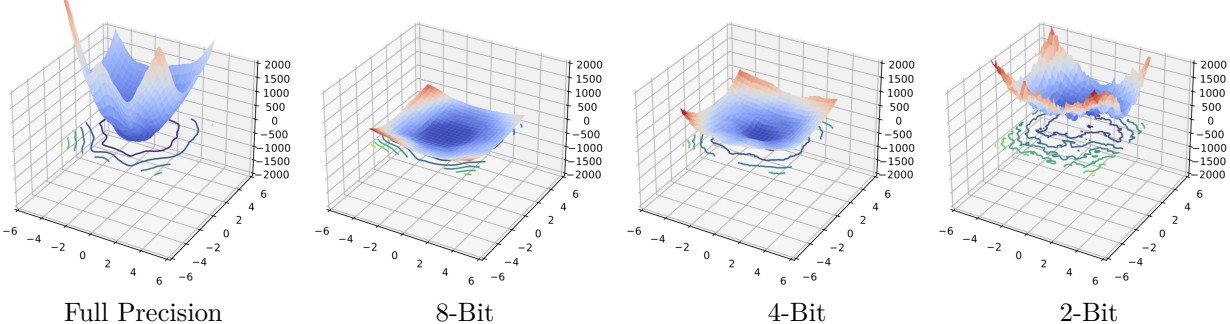

Figure 2: Visualization of the loss landscape for the full precision and quantized ResNet-18 models trained on CIFAR-10. It is observed that 8-bit quantized model has the most flat minima compared to other models. Additionally, as the quantization precision decreases, the flatness of the loss landscape diminishes.

### 4.1 Flatness of Minima and Generalization

In this section, we conduct experiments that demonstrate quantized neural networks enjoy better flatness in their loss landscape compared to their full-precision counterparts; this finding is contrary to the assumption of some of the recent studies Liu et al. (2021); Wang et al. (2022). In those studies, it is assumed that quantization results in sharper minima. We believe that the root of this assumption might be that the authors of those works have not considered the magnitude of the network weights in measuring the sharpness. However, as Jiang et al. (2019) and Dziugaite et al. (2020) show, the flatness measures that take the magnitude of the parameters into account Keskar et al. (2016), are better indicators of generalization.

As Table 2 shows, for a given backbone, the magnitude of the network parameters (calculated as the $L2$-norm of weights) are very different among different quantization levels; therefore, simply measuring the loss landscape flatness using sharpness or PAC-Bayesian bounds without considering the magnitude of the weights could be misleading.

To capture the flatness around the local minima of a given network $f(\boldsymbol{w})$, we utilize the PAC-Bayesian bounds McAllester (1999) and sharpness measures Keskar et al. (2016). The former adds Gaussian perturbations to the network parameters and captures the average (expected) flatness within a bound, while the latter captures the worst-case flatness, i.e. sharpness, through adding adversarial worst-case perturbations to the parameters. We use the same formulation and implementation as specified by Jiang et al. (2019) and Dziugaite et al. (2020); in particular, similar to their approach, we measure and report these metrics by considering the trained model and the network-at-initialization parameters as the *origin* and *initialization* tensors, respectively. Moreover, as discussed in the above paragraph, and indicated in Table 2, the magnitude-aware versions of these metrics are the most reliable way of capturing the flatness of a network, hence we also report the magnitude-aware measurements, and they will be the main measure of loss landscape flatness. Details about these metrics and their formulation are in the supplementary Section A.

As shown in Table 3, for the 3 datasets of CIFAR-10, CIFAR-100 , and ImageNet-1k, and over a variety of network backbones, the quantized models enjoy flatter loss landscapes which is an indicator of better generalization to unseen data. An important observation from the experiments reported in Table 3 is that relying solely on sharpness or PAC-Bayesian measures without considering the magnitude of the network parameters might create the assumption that quantization does increase the network sharpness. We suspect that this might have indeed been the cause of this assumption in the works of Liu et al. (2021); Wang et al. (2022) which assume worse sharpness for quantized models and then propose Sharpness-Aware-Minimization (SAM) coupled with Quantization-Aware-Training (QAT). However, our empirical studies demonstrate that when the magnitude of the parameters is taken into account, quantization does actually improve flatness, and the finding that SAM can help quantized models achieve further flatness does not necessarily mean that quantized models have sharper minima compared to the non-quantized counterparts.

Table 3: Sharpness-based measures for different quantization levels over 3 image datasets. The lowest values in each category correspond to the lowest complexity that is considered to be the best (indicated by ■ ). All the values are normalized by calculating $\sqrt{\frac{x}{m}}$ where $x$ is the value of the measure and $m$ is the size of the dataset.

| Dataset | Model | Precision | PAC-Bayesian | | | | Sharpness | | | |
|---|---|---|---|---|---|---|---|---|---|---|
| | | | Init | Orig | Mag-Init | Mag-Orig | Init | Orig | Mag-Init | Mag-Orig |
| CIFAR10 | NiN (4x10) | FP32 | 2.264 | 2.2 | 7.635 | 7.594 | 0.589 | 0.572 | 8.219 | 8.181 |
| | | Int8 | 4.204 | 3.626 | 6.435 | 6.176 | 0.292 | 0.252 | 6.853 | 6.610 |
| | | Int4 | 2.482 | 2.143 | 6.419 | 6.162 | 1.444 | 1.247 | 6.826 | 6.586 |
| | | Int2 | 1.588 | 1.32 | 6.171 | 5.833 | 1.152 | 0.958 | 6.454 | 6.131 |
| | NiN (4x12) | FP32 | 1.469 | 1.328 | 7.974 | 7.770 | 0.359 | 0.324 | 9.216 | 9.040 |
| | | Int8 | 6.057 | 4.866 | 7.718 | 7.256 | 0.259 | 0.208 | 8.655 | 8.245 |
| | | Int4 | 2.658 | 2.131 | 7.765 | 7.302 | 1.335 | 1.07 | 8.56 | 8.142 |
| | | Int2 | 1.918 | 1.493 | 7.654 | 7.119 | 0.781 | 0.608 | 8.513 | 8.034 |
| | ResNet-18 | FP32 | 1.186 | 1.135 | 3.659 | 3.617 | 1.447 | 1.383 | 4.399 | 4.364 |
| | | Int8 | 0.893 | 0.834 | 3.355 | 3.285 | 0.426 | 0.398 | 4.112 | 4.055 |
| | | Int4 | 1.786 | 1.673 | 3.291 | 3.223 | 0.433 | 0.405 | 4.037 | 3.981 |
| | | Int2 | 1.368 | 1.267 | 3.238 | 3.156 | 0.819 | 0.759 | 4.074 | 4.012 |
| | ResNet-50 | FP32 | 1.803 | 1.647 | 5.304 | 5.193 | 1.237 | 1.13 | 6.303 | 6.210 |
| | | Int8 | 3.911 | 2.901 | 4.729 | 4.300 | 0.988 | 0.733 | 5.472 | 5.106 |
| | | Int4 | 8.937 | 8.793 | 6.394 | 6.377 | 3.71 | 3.65 | 6.684 | 6.669 |
| | | Int2 | 1.991 | 1.431 | 4.638 | 4.151 | 1.926 | 1.385 | 5.499 | 5.094 |
| CIFAR100 | NiN (5x10) | FP32 | 4.266 | 4.192 | 9.33 | 9.302 | 0.859 | 0.844 | 10.467 | 10.443 |
| | | Int8 | 7.354 | 6.339 | 7.451 | 7.155 | 0.474 | 0.409 | 8.084 | 7.812 |
| | | Int4 | 3.101 | 2.673 | 7.399 | 7.101 | 0.473 | 0.408 | 8.032 | 7.759 |
| | | Int2 | 2.25 | 1.939 | 6.138 | 5.776 | 0.313 | 0.27 | 7.774 | 7.491 |
| | NiN (5x12) | FP32 | 3.505 | 3.409 | 10.958 | 10.904 | 0.777 | 0.755 | 12.041 | 11.992 |
| | | Int8 | 1.712 | 1.561 | 9.175 | 8.963 | 0.582 | 0.531 | 9.956 | 9.761 |
| | | Int4 | 3.934 | 3.595 | 9.274 | 9.069 | 0.581 | 0.531 | 9.794 | 9.599 |
| | | Int2 | 4.343 | 3.922 | 9.479 | 9.252 | 0.557 | 0.503 | 9.828 | 9.609 |
| | ResNet-18 | FP32 | 3.535 | 3.495 | 4.243 | 4.234 | 3.429 | 3.39 | 4.795 | 4.786 |
| | | Int8 | 6.031 | 5.696 | 3.685 | 3.631 | 1.194 | 1.128 | 4.232 | 4.185 |
| | | Int4 | 2.381 | 2.25 | 3.591 | 3.536 | 1.166 | 1.102 | 4.117 | 4.069 |
| | | Int2 | 3.704 | 3.443 | 3.538 | 3.465 | 27.983 | 27.247 | 4.65 | 4.611 |
| | ResNet-50 | FP32 | 4.396 | 4.265 | 5.918 | 5.883 | 4.732 | 4.591 | 6.797 | 6.768 |
| | | Int8 | 5.583 | 4.279 | 4.775 | 4.385 | 2.445 | 1.874 | 5.613 | 5.285 |
| | | Int4 | 3.076 | 2.397 | 5.273 | 4.945 | 2.386 | 1.859 | 6.809 | 6.558 |
| | | Int2 | 29.727 | 29.531 | 5.253 | 5.247 | 37.893 | 38.124 | 8.343 | 8.339 |
| ImageNet-1K | ResNet-18 | FP32 | 11.694 | 11.584 | 12.378 | 12.355 | 349.235 | 345.962 | 20.069 | 20.055 |
| | | Int8 | 7.836 | 5.303 | 10.1 | 8.902 | 104.91 | 70.994 | 18.416 | 17.786 |
| | | Int4 | 4.615 | 3.108 | 10.072 | 8.853 | 104.557 | 70.419 | 18.41 | 17.772 |
| | | Int2 | 16.397 | 10.563 | 11.004 | 9.770 | 101.31 | 65.266 | 18.362 | 17.649 |
| | ResNet-50 | FP32 | 7.942 | 7.144 | 22.163 | 21.826 | 5.067 | 4.556 | 27.746 | 27.418 |
| | | Int8 | 20.398 | 14.344 | 17.597 | 16.272 | 11.208 | 7.881 | 20.104 | 18.995 |
| | | Int4 | 35.011 | 24.637 | 17.809 | 16.503 | 258.118 | 181.636 | 19.162 | 18.833 |
| | | Int2 | 245.654 | 173.287 | 17.954 | 17.023 | 258.722 | 182.505 | 24.051 | 24.007 |
| | DeiT-T | FP32 | 8.653 | 8.123 | 19.651 | 18.226 | 7.017 | 6.753 | 31.924 | 31.5 |
| | | Int8 | 26.544 | 27.352 | 18.232 | 17.563 | 7.445 | 5.126 | 22.524 | 23.432 |
| | | Int4 | 35.786 | 33.982 | 17.983 | 16.148 | 5.927 | 4.672 | 20.122 | 23.765 |
| | | Int2 | 236.322 | 171.234 | 19.865 | 18.982 | 218.621 | 169.972 | 32.114 | 33.763 |

### 4.1.1 Loss Landscape Visualization

The loss landscape of quantized neural networks can be effectively visualized. Using the technique outlined in Li et al. (2018), we projected the loss landscape of quantized models and the full precision ResNet-18 models trained on the CIFAR-10 dataset onto a three-dimensional plane. The visual representation, as illustrated in Figure 2, clearly demonstrates that the loss landscape associated with quantized models is comparatively flatter. This observation confirms the findings presented in Table 3.

### 4.2 Measuring the Generalization Gap

To study the generalization behaviors of quantized models, we have trained almost 2000 models on the CIFAR-10, CIFAR-100 and ImageNet-1K datasets. Our goal is to approximate the $\mathcal{L}_{\mathcal{D}}(\boldsymbol{w}) - \mathcal{L}_{\mathcal{S}}(\boldsymbol{w})$, i.e., the generalization gap, by utilizing the data that is unseen during the training process (the test data). Without loss of generality, herein, we will refer to the difference between test data loss and training data loss as the generalization gap.

Following the guidelines of Jiang et al. (2019) and Dziugaite et al. (2020) to remove the effect of randomness from our analysis of generalization behavior, for smaller datasets (CIFAR-10 and CIFAR100), we construct a pool of trained models by varying 5 commonly used hyperparameters over the fully convolutional "Network-in-Network" architecture Lin et al. (2013). The hyperparameter list includes learning rate, weight decay, optimization algorithm, architecture depth, and layer width. In our experiments, each hyperparameter has 3 choices; therefore, the number of trained models per quantization level is $3^5 = 243$, with the number of bits considered being selected from the following values: 2, 4, and 8, and the resulting models are compared with their full-precision counterpart. Thus, in total, we will have $4 \times 243 = 992$ models trained per dataset, over CIFAR-10 and CIFAR-100 datasets, which gives us almost 2000 trained models. For more details regarding hyperparameter choices and model specifications, please refer to the supplementary material Section B. Lastly, for ImageNet-1k, we measured the generalization gap on both CNN and ViT models.

In Jiang et al. (2019), to measure the generalization gap of a model, the authors first train the model until the training loss converges to a threshold (0.01). Here, we argue that this approach might not be optimal when quantization enters the picture. First, lower bit-resolution quantized models have lower learning capacity compared to the higher bit-resolution quantized or the full-precision ones; our proof in Equation 7 also indicates that the learning capabilities of a given network diminish as the number of quantization bits decreases. Second, early stopping of the training process may hinder the trained models from appropriately converging to flatter local minima, which quantized models enjoy in their loss landscape. Therefore, we apply a different training approach. Each model is trained for 300 epochs by lowering the learning rate by a factor of 10 at epochs 100 and 200, and at the end, the model corresponding to the lowest training loss is chosen.

Table 1 summarizes the results of these experiments. The accuracy-generalization trade-off is demonstrated through these experiments. The training loss and training accuracy of lower-resolution quantized models are negatively impacted. However, they enjoy better generalization. Some additional interesting results can be inferred from Table 1. Notably, 8-bit quantization is almost on par with the full-precision counterpart on all the metrics. This is also evident in Table 3, where we studied the sharpness-based measures. The other interesting observation is that although training losses vary among the models, the test loss is almost the same among all; this, in turn, indicates that full-precision and high-resolution quantized models have a higher degree of overfitting, which could result from converging to sharper local minima.

### 4.3 Generalization Under Distorted Data

In addition to assessing the generalization metrics outlined in the previous sections, we sought to investigate some real-world implications that the generalization quality of the quantized models would have. To this end, we evaluate the performance of vision models in the case when the input images are distorted or corrupted by common types of perturbations. We take advantage of the comprehensive benchmark provided in Hendrycks & Dietterich (2019) where they identify 15 types of common distortions and measure the performance of the models under different levels of severity for each distortion. Table 4 presents the generalization gaps as calculated by the difference of loss on the corrupted dataset and the loss on training data for 5 levels of severity for ResNet-18 and ResNet-50 trained on ImageNet-1K. By augmenting the test dataset in this way, we are unlocking more unseen data for evaluating the generalization of our models. As is evident through these experiments, quantized models maintain their superior generalization under most of the distortions. Accuracies of the models on the distorted dataset, as well as results and discussions on more architectures and datasets and details on conducting our experiments, are available in the supplementary material Section C.

Table 4: Effect of distortion on generalization gap with quantized models. Compared to FP32 column, we have highlighted better generalization gap with 🟩 and 🟥 to show the opposite.

| Model | Augmentation | Severity 1 | | | | Severity 2 | | | | Severity 3 | | | | Severity 4 | | | | Severity 5 | | | |
|---|---|---|---|---|---|---|---|---|---|---|---|---|---|---|---|---|---|---|---|---|---|
| | | FP32 | Int8 | Int4 | Int2 | FP32 | Int8 | Int4 | Int2 | FP32 | Int8 | Int4 | Int2 | FP32 | Int8 | Int4 | Int2 | FP32 | Int8 | Int4 | Int2 |
| ResNet-18 | Gaussian Noise | 1.067 | 0.86 | 0.939 | 1.21 | 1.913 | 1.46 | 1.629 | 2.201 | 3.439 | 2.52 | 2.796 | 3.658 | 5.529 | 3.96 | 4.352 | 5.115 | 7.653 | 5.62 | 6.117 | 6.093 |
| | Shot Noise | 1.238 | 0.96 | 1.054 | 1.333 | 2.289 | 1.72 | 1.887 | 2.421 | 3.801 | 2.75 | 2.987 | 3.685 | 6.438 | 4.4 | 4.739 | 5.304 | 7.919 | 5.41 | 5.805 | 6.003 |
| | Impulse Noise | 2.235 | 1.78 | 2.058 | 2.324 | 3.177 | 2.35 | 2.636 | 3.279 | 4.061 | 2.9 | 3.185 | 4.001 | 6.096 | 4.26 | 4.595 | 5.324 | 7.781 | 5.57 | 5.995 | 6.114 |
| | Defocus Noise | 0.979 | 0.89 | 0.858 | 0.822 | 1.432 | 1.37 | 1.325 | 1.302 | 2.394 | 2.34 | 2.263 | 2.217 | 3.285 | 3.19 | 3.099 | 2.934 | 3.983 | 3.92 | 3.808 | 3.498 |
| | Glass Blue | 1.18 | 1.07 | 1.031 | 0.985 | 1.969 | 1.86 | 1.812 | 1.804 | 3.822 | 3.83 | 3.727 | 3.54 | 4.221 | 4.25 | 4.15 | 3.909 | 4.652 | 4.62 | 4.542 | 4.139 |
| | Motion Blur | 0.687 | 0.55 | 0.542 | 0.509 | 1.37 | 1.22 | 1.245 | 1.261 | 2.521 | 2.42 | 2.454 | 2.381 | 3.7 | 3.64 | 3.678 | 3.412 | 4.285 | 4.23 | 4.275 | 3.875 |
| | Zoom Blur | 1.518 | 1.38 | 1.382 | 1.386 | 2.219 | 2.1 | 2.119 | 2.094 | 2.688 | 2.58 | 2.588 | 2.518 | 3.213 | 3.12 | 3.137 | 3.028 | 3.666 | 3.59 | 3.599 | 3.437 |
| | Snow | 1.401 | 1.01 | 0.998 | 1.124 | 3.215 | 2.36 | 2.374 | 2.643 | 2.969 | 2.07 | 2.094 | 2.315 | 3.97 | 2.81 | 2.869 | 3.074 | 4.515 | 3.48 | 3.517 | 3.572 |
| | Frost | 0.949 | 0.66 | 0.626 | 0.633 | 2.093 | 1.68 | 1.681 | 1.812 | 2.978 | 2.53 | 2.553 | 2.688 | 3.141 | 2.74 | 2.766 | 2.893 | 3.713 | 3.31 | 3.362 | 3.447 |
| | Fog | 0.809 | 0.42 | 0.444 | 0.405 | 1.214 | 0.68 | 0.734 | 0.774 | 1.857 | 1.18 | 1.273 | 1.431 | 2.347 | 1.68 | 1.762 | 1.958 | 3.77 | 3.03 | 3.141 | 3.275 |
| | Brightness | 0.121 | 0.04 | 0.019 | 0.155 | 0.221 | 0.1 | 0.08 | 0.062 | 0.378 | 0.19 | 0.184 | 0.084 | 0.631 | 0.37 | 0.36 | 0.323 | 0.986 | 0.62 | 0.626 | 0.672 |
| | Contrast | 0.523 | 0.24 | 0.232 | 0.13 | 0.867 | 0.4 | 0.413 | 0.396 | 1.627 | 0.81 | 0.861 | 1.031 | 3.61 | 2.37 | 2.529 | 2.921 | 5.264 | 4.63 | 4.765 | 4.479 |
| | Elastic | 0.538 | 0.43 | 0.406 | 0.287 | 2.026 | 1.95 | 1.911 | 1.833 | 1.116 | 1.03 | 0.969 | 0.884 | 1.997 | 1.94 | 1.844 | 1.755 | 4.112 | 4.11 | 3.957 | 3.57 |
| | Pixelate | 0.612 | 0.5 | 0.492 | 0.416 | 0.599 | 0.51 | 0.506 | 0.465 | 1.889 | 1.72 | 1.734 | 1.958 | 3.046 | 2.93 | 2.88 | 3.306 | 3.369 | 3.32 | 3.313 | 3.51 |
| | JPEG | 0.59 | 0.48 | 0.468 | 0.375 | 0.801 | 0.68 | 0.674 | 0.627 | 0.972 | 0.85 | 0.841 | 0.824 | 1.599 | 1.46 | 1.446 | 1.491 | 2.615 | 2.43 | 2.405 | 2.487 |
| MoblieNet V2 | Gaussian Noise | 1.041 | 0.76 | 0.857 | 2.78 | 1.923 | 1.382 | 1.536 | 3.755 | 3.425 | 2.5 | 2.762 | 5.009 | 5.251 | 4.065 | 4.518 | 6.182 | 6.997 | 5.815 | 6.423 | 7.124 |
| | Shot Noise | 1.132 | 0.843 | 1.027 | 2.926 | 2.214 | 1.591 | 1.846 | 3.975 | 3.67 | 2.624 | 3.013 | 5.058 | 5.891 | 4.363 | 5.011 | 6.36 | 7.045 | 5.418 | 6.137 | 6.96 |
| | Impulse Noise | 1.635 | 1.483 | 1.585 | 3.043 | 2.597 | 2.223 | 2.284 | 4.144 | 3.423 | 2.751 | 2.901 | 4.961 | 5.302 | 4.171 | 4.595 | 6.296 | 6.979 | 5.753 | 6.315 | 7.126 |
| | Defocus Noise | 0.863 | 0.799 | 1.005 | 3.856 | 1.326 | 1.266 | 1.519 | 4.286 | 2.23 | 2.213 | 2.434 | 4.858 | 3.059 | 2.983 | 3.328 | 5.119 | 3.784 | 3.655 | 4.2 | 5.303 |
| | Glass Blue | 1.223 | 1.141 | 1.509 | 3.538 | 2.115 | 2.039 | 2.443 | 4.257 | 4.01 | 4.003 | 4.309 | 4.943 | 4.375 | 4.353 | 4.564 | 5.039 | 4.668 | 4.601 | 4.709 | 5.141 |
| | Motion Blur | 0.643 | 0.53 | 0.641 | 3.068 | 1.335 | 1.209 | 1.354 | 3.768 | 2.392 | 2.282 | 2.435 | 4.349 | 3.5 | 3.418 | 3.588 | 4.774 | 4.108 | 4.028 | 4.235 | 4.983 |
| | Zoom Blur | 1.539 | 1.423 | 1.607 | 3.564 | 2.282 | 2.185 | 2.358 | 3.97 | 2.774 | 2.685 | 2.886 | 4.335 | 3.317 | 3.263 | 3.492 | 4.567 | 3.797 | 3.738 | 4.035 | 4.817 |
| | Snow | 1.253 | 0.904 | 1.168 | 2.377 | 3.074 | 2.429 | 2.694 | 4.017 | 2.838 | 2.16 | 2.497 | 3.869 | 3.775 | 2.945 | 3.286 | 4.797 | 4.562 | 3.776 | 3.996 | 5.151 |
| | Frost | 0.941 | 0.658 | 0.8 | 2.236 | 2.193 | 1.784 | 2.021 | 3.713 | 3.154 | 2.673 | 2.969 | 4.682 | 3.345 | 2.904 | 3.223 | 4.94 | 3.956 | 3.484 | 3.835 | 5.46 |
| | Fog | 0.699 | 0.354 | 0.822 | 3.874 | 1.084 | 0.624 | 1.24 | 4.454 | 1.715 | 1.145 | 1.802 | 4.929 | 2.256 | 1.675 | 2.111 | 4.969 | 3.792 | 3.116 | 3.298 | 5.371 |
| | Brightness | 0.034 | 0.05 | 0.019 | 1.342 | 0.143 | 0.008 | 0.089 | 1.359 | 0.303 | 0.17 | 0.211 | 1.549 | 0.595 | 0.303 | 0.422 | 1.987 | 1.002 | 0.591 | 0.752 | 2.663 |
| | Contrast | 0.482 | 0.188 | 0.816 | 3.717 | 0.846 | 0.394 | 1.513 | 4.571 | 1.624 | 0.899 | 3.095 | 5.556 | 3.66 | 2.725 | 5.816 | 6.396 | 5.411 | 4.881 | 6.505 | 6.59 |
| | Elastic | 0.442 | 0.347 | 0.481 | 2.396 | 1.973 | 1.871 | 2.105 | 4.012 | 0.962 | 0.87 | 1.113 | 2.474 | 1.913 | 1.807 | 2.21 | 2.959 | 4.106 | 3.982 | 4.693 | 4.036 |
| | Pixelate | 0.926 | 0.653 | 0.872 | 1.883 | 1.444 | 1.02 | 0.934 | 1.838 | 2.155 | 1.822 | 2.468 | 2.172 | 3.111 | 3.064 | 3.773 | 2.755 | 3.979 | 3.993 | 3.988 | 3.301 |
| | JPEG | 0.491 | 0.382 | 0.554 | 1.754 | 0.675 | 0.552 | 0.784 | 1.848 | 0.826 | 0.693 | 0.967 | 1.928 | 1.357 | 1.165 | 1.555 | 2.182 | 2.182 | 1.902 | 2.462 | 2.545 |
| ResNet-50 | Gaussian Noise | 0.938 | 0.914 | 0.928 | 0.973 | 1.437 | 1.282 | 1.112 | 1.571 | 2.363 | 2.047 | 2.513 | 2.89 | 3.719 | 3.255 | 3.754 | 4.88 | 5.134 | 4.999 | 5.339 | 7.828 |
| | Shot Noise | 0.961 | 0.946 | 0.957 | 1.026 | 1.585 | 1.408 | 1.246 | 1.83 | 2.448 | 2.166 | 2.023 | 3.215 | 4.084 | 3.697 | 3.887 | 5.748 | 4.924 | 4.919 | 5.229 | 7.656 |
| | Impulse Noise | 1.703 | 1.652 | 1.676 | 1.789 | 2.013 | 1.874 | 1.464 | 2.373 | 2.564 | 2.295 | 1.995 | 3.211 | 3.962 | 3.507 | 3.458 | 5.435 | 5.28 | 4.942 | 5.105 | 8.123 |
| | Defocus Noise | 1.059 | 1.042 | 0.911 | 0.869 | 1.441 | 1.414 | 1.309 | 1.298 | 2.344 | 2.311 | 2.286 | 2.231 | 3.244 | 3.225 | 3.226 | 3.164 | 4.052 | 4.049 | 4.019 | 3.994 |
| | Glass Blue | 1.349 | 1.27 | 1.011 | 1.457 | 2.297 | 2.169 | 1.829 | 2.088 | 4.613 | 4.551 | 4.185 | 4.346 | 5.057 | 5.009 | 4.815 | 4.793 | 5.399 | 5.376 | 5.34 | 5.102 |
| | Motion Blur | 0.731 | 0.638 | 0.623 | 0.578 | 1.314 | 1.307 | 1.19 | 1.238 | 2.563 | 2.551 | 2.337 | 2.501 | 4.148 | 4.057 | 3.672 | 3.96 | 5.048 | 5.033 | 4.404 | 4.75 |
| | Zoom Blur | 1.509 | 1.473 | 1.261 | 1.361 | 2.252 | 2.187 | 2.037 | 2.134 | 2.822 | 2.736 | 2.571 | 2.697 | 3.44 | 3.337 | 3.139 | 3.325 | 4.039 | 3.949 | 3.676 | 3.916 |
| | Snow | 1.229 | 1.13 | 1.048 | 1.143 | 2.62 | 2.529 | 2.481 | 2.933 | 2.375 | 2.317 | 2.193 | 2.478 | 3.127 | 3.016 | 2.941 | 3.347 | 3.697 | 3.437 | 3.8 | 4.209 |
| | Frost | 0.845 | 0.837 | 0.674 | 0.653 | 1.769 | 1.726 | 1.506 | 1.785 | 2.563 | 2.522 | 2.245 | 2.588 | 2.761 | 2.72 | 2.435 | 2.816 | 3.322 | 3.291 | 2.972 | 3.427 |
| | Fog | 0.691 | 0.685 | 0.582 | 0.501 | 0.897 | 0.876 | 0.784 | 0.926 | 1.305 | 1.248 | 1.167 | 1.337 | 1.81 | 1.657 | 1.635 | 1.909 | 3.261 | 2.96 | 2.979 | 3.569 |
| | Brightness | 0.345 | 0.25 | 0.21 | 0.081 | 0.382 | 0.314 | 0.259 | 0.431 | 0.453 | 0.446 | 0.341 | 0.222 | 0.584 | 0.514 | 0.473 | 0.367 | 0.787 | 0.716 | 0.674 | 0.609 |
| | Contrast | 0.545 | 0.449 | 0.42 | 0.308 | 0.69 | 0.677 | 0.568 | 0.494 | 1.047 | 0.998 | 0.867 | 1.051 | 2.387 | 2.173 | 1.85 | 2.624 | 4.686 | 4.394 | 4.393 | 4.914 |
| | Elastic | 0.655 | 0.625 | 0.517 | 0.422 | 2.274 | 2.243 | 1.908 | 2.122 | 1.602 | 1.571 | 1.242 | 1.416 | 2.704 | 2.671 | 2.209 | 2.591 | 5.598 | 5.522 | 4.647 | 5.348 |
| | Pixelate | 0.871 | 0.71 | 0.727 | 0.684 | 1.042 | 1.021 | 1.015 | 0.778 | 1.971 | 1.575 | 1.556 | 1.974 | 3.373 | 3.229 | 3.208 | 3.431 | 4.038 | 4.014 | 3.996 | 4.179 |
| | JPEG | 0.793 | 0.701 | 0.67 | 0.522 | 0.957 | 0.98 | 0.832 | 0.704 | 1.092 | 1.023 | 0.96 | 0.849 | 1.537 | 1.425 | 1.356 | 1.394 | 2.236 | 2.118 | 1.967 | 2.228 |

## 5   Conclusion

In this work, we investigated the generalization properties of quantized neural networks, which have received limited attention despite their significant impact on model performance. We demonstrated that quantization has a regularization effect and it leads to improved generalization capabilities. We empirically show that quantization could facilitate the convergence of models to flatter minima. Lastly, on distorted data, we provided empirical evidence that quantized models exhibit improved generalization compared to their full-precision counterparts across various experimental setups. Through the findings of this study, we hope that the inherent generalization capabilities of quantized models can be used to further improve their performance.

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

## A    Flatness Landscape

The PAC-Bayesian and sharpness generalization measures both make use of the PAC-Bayes bounds, which estimate the bounds of the generalization error of a predictor (i.e. neural network). In our case, the PAC-Bayes bound is a function of the KL divergence of the prior distribution and posterior distribution of the model parameters, where the prior distribution is drawn without knowledge of the dataset and the posterior distribution is a perturbation on the trained parameters. It has been shown that when both distributions are isotropic Gaussian distributions, then PAC-Bayesian bounds are a good measure of generalization in small-scale experiments. We refer the reader to Jiang et al. (2019) for more detailed analysis and derivations, which we summarize here. The PAC-Bayes generalization measures are defined below:

$$\mu_{\text{pac-bayes-init}}(f_{\boldsymbol{w}})) = \frac{||\boldsymbol{w} - \boldsymbol{w}^0||_2^2}{4\sigma^2} + \log(\frac{m}{\sigma}) + 10 \tag{15}$$

$$\mu_{\text{pac-bayes-orig}}(f_{\boldsymbol{w}})) = \frac{||\boldsymbol{w}||_2^2}{4\sigma^2} + \log(\frac{m}{\delta}) + 10 \tag{16}$$

Where $\sigma$ is chosen to be the largest number such that $\mathbb{E}_{\boldsymbol{u} \sim \mathcal{N}(\mu, \sigma^2 I)}[\hat{\mathcal{L}}(f_{\boldsymbol{w}+\boldsymbol{u}})] \leq 0.1$, and $m$ is the sample size of the dataset

From the same PAC-Bayesian bound framework, we can also derive the sharpness measure, by using the worst-case noise $\alpha$ rather than the Gaussian sampled noise.

$$\mu_{\text{sharpness-init}}(f_{\boldsymbol{w}})) = \frac{||\boldsymbol{w} - \boldsymbol{w}^0||_2^2 \log(2\omega)}{4\alpha^2} + \log(\frac{m}{\sigma}) + 10 \tag{17}$$

$$\mu_{\text{sharpness-orig}}(f_{\boldsymbol{w}})) = \frac{||\boldsymbol{w}||_2^2 \log(2\omega)}{4\alpha^2} + \log(\frac{m}{\delta}) + 10 \tag{18}$$

Where $\alpha$ is chosen to be the largest number such that $\max_{|u_i| \leq \alpha} \hat{\mathcal{L}}(f_{\boldsymbol{w}+\boldsymbol{u}}) \leq 0.1$ and $\omega$ is the number of parameters in the model.

For magnitude-aware measures Keskar et al. (2016), the ratio of the magnitude of the perturbation to the magnitude of the parameter is bound by a constant $\alpha'$. By bounding the ratio of perturbation to parameter magnitude, we prevent parameters from changing signs. This change leads to the following magnitude-aware generalization measures:

$$\mu_{pac-bayes-mag-init}(f_{\boldsymbol{w}}) = \frac{1}{4} \sum_{i=1}^{\omega} \log\left(\frac{\epsilon^2 + (\sigma'^2 + 1)||\boldsymbol{w} - \boldsymbol{w}^0||_2^2/\omega}{\epsilon^2 + \sigma'^2|w_i - w_i^0|^2}\right) + \log(\frac{m}{\delta}) + 10 \tag{19}$$

$$\mu_{pac-bayes-mag-orig}(f_{\boldsymbol{w}}) = \frac{1}{4} \sum_{i=1}^{\omega} \log\left(\frac{\epsilon^2 + (\sigma'^2 + 1)||\boldsymbol{w}||_2^2/\omega}{\epsilon^2 + \sigma'^2|w_i - w_i^0|^2}\right) + \log(\frac{m}{\delta}) + 10 \tag{20}$$

$$\mu_{sharpness-mag-init}(f_{\boldsymbol{w}}) = \frac{1}{4} \sum_{i=1}^{\omega} \log\left(\frac{\epsilon^2 + (\alpha'^2 + 4\log(2\omega/\delta))||\boldsymbol{w} - \boldsymbol{w}^0||_2^2/\omega}{\epsilon^2 + \alpha'^2|w_i - w_i^0|^2}\right) + \log(\frac{m}{\delta}) + 10 \tag{21}$$

$$\mu_{sharpness-mag-orig}(f_{\boldsymbol{w}}) = \frac{1}{4} \sum_{i=1}^{\omega} \log\left(\frac{\epsilon^2 + (\alpha'^2 + 4\log(2\omega/\delta))||\boldsymbol{w}||_2^2/\omega}{\epsilon^2 + \alpha'^2|w_i - w_i^0|^2}\right) + \log(\frac{m}{\delta}) + 10 \tag{22}$$

Where $\epsilon = 0.001$ and $\sigma$ is chosen to be the largest number such that $\mathbb{E}_{\boldsymbol{u}}\left[\hat{\mathcal{L}}(f_{\boldsymbol{w}+\boldsymbol{u}})\right] \leq 0.1$,

## B    Experiment Setup For Measuring Sharpness-based Metrics

### B.1    Training Setup

We used different models and datasets to compute the generalization gap using proxy metrics described in Section A.

Our experiments employed the LSQ method Esser et al. (2020) for weight quantization. The CIFAR-10, CIFAR-100, and ImageNet datasets were utilized for testing purposes. We applied three distinct quantization levels for quantized models: 2, 4, and 8 bits. The CIFAR-10 and CIFAR-100 NiN models are trained with a base width of 25, and they are trained for 300 epochs, with an SGD optimizer, an initial learning rate of 0.1, momentum of 0.9, and a weight decay of 0.0001. We utilize a multi-step scheduler with steps at epochs 100 and 200, and the gamma is 0.1. The ResNet models that we use for these two datasets have a base width of 16 and use the same optimizer as the NiN network. However, these models are trained for 200 epochs, and the steps happen at epochs 80 and 160. The ResNet models we utilize for comparing sharpness-based measures for the ImageNet dataset have a base width of 64. We again use the same optimizer only with a different learning rate of 0.01. We fine-tune the models from Pytorch pre-trained weights for 120 epochs, and the steps happen at epochs 30, 60, and 90.

## B.2    Measuring the Metrics

To measure the PAC-Bayesian and sharpness measures, we measure these metrics for the cases of magnitude aware and the normal for each quantization level. In each case, we run the search for finding the maximum amount of possible noise ($\sigma$), for 15 iterations, and within each iteration we calculate the mean of the accuracy on the training data over 10 runs to remove the effect of randomness. As an additional step in calculating the sharpness measures, we perform the gradient ascent step to maximize the loss value for 20 iterations. We use a learning rate of 0.0001 with an SGD optimizer for the gradient ascent process.

## B.3    Measuring Generalization Gaps

In our experiments for measuring the generalization gaps, we trained almost 2000 CIFAR-10 and CIFAR-100 models. The main backbone in all these experiments was NiN. We trained the models over the variation of hyperparameter values for 5 hyperparameter, and each hyperparameter had 3 choices. For the case of CIFAR-10, here are the values for hyperparameters:

- Optimizer algorithm: {SGD, ADAM, RMSProp}

- Learning rate: {0.1, 0.05, 0.01} for SGD, {0.001, 0.0005, 0.0001} for ADAM and RMSProp

- Weight decay: {0.0, 0.0001, 0.0002}

- Width multiplier: {8, 10, 12}

- Depth multiplier: {2, 3, 4}

For CIFAR-100 everything is the same with the minor difference of depth multipliers being in the set of {3, 4, 5}.

Each NiN training instance is trained for 300 epochs, in every case a step scheduler with steps at the 100th and 200th epoch and a gamma of 0.1 is utilized. The model with the lowest loss on training data is used with no information about the test data. Then the statistics in 1 are generated.

## B.4    Computation Requirements

To train the NiN models for each quantization level, we use one NVIDIA A100 GPU with a batch size of 128. Each experiment takes almost 6 days to run, which on average is equivalent to 35 minutes per model training. We use 8 GPUs, 4 for CIFAR-10 and 4 for CIFAR-100.

For evaluating the sharpness measures, the main bottleneck is for ImageNet models, as evaluating the sharpness measures for each quantization level requires almost 600 evaluations on the training data in the worst. Running each quantization level on one NVIDIA A100 GPU requires 33 hours on average.

## C   Distortion Experiments

These are the extended results for investigating the generalization gap under distortion. We provide a generalization gap of quantized and full precision models on augmented datasets.

### C.1   Training Setup

For full precision models, we used pre-trained models publicly available on the Pytorch website Pyt. For quantized models, we use weight quantization using LSQ Esser et al. (2020) method. We use CIFAR-100 and ImageNet datasets in our tests. We use three different quantization levels for quantized models: 2, 4, and 8 bits. We use a multi-step scheduler with steps at 30, 60, and 90 with an initial learning rate of 0.01 and gamma of 0.1. We use weight decay of 1e-4 and SGD optimizer. We trained all models for 120 epochs. Finally, we used pre-trained models from Pytorch to initialize weights for LSQ quantization.

### C.2   Data Preparation

For augmented datasets, we use the corrupted Imagenet-C and CIFAR100-C datasets proposed in Hendrycks & Dietterich (2019). Table 6 presents the results of the experiments performed on the ResNet-18, MoobileNet V2 and ResNet-50 models trained on the ImageNet dataset, and Table 5 present the results for ResNet-18, MobileNet V1, and VGG-19 models on CIFAR-100. These tables show the effect of distortion on the generalization gap of quantized models when various types and severity levels of distortions are used. Specifically, 15 different types of distortions were applied to the models. For each distortion type, the generalization gap was computed by subtracting the test loss on the distorted dataset from the loss on the original ImageNet training dataset.

### C.3   Computation Setup

For these experiments, we used 8 NVIDIA A100 GPUs with 40 GB of RAM to train ImageNeta and CIFAR-100 models. With the above training hardware and submitted code, each ImageNet model takes 18 hours to train on average. CIFAR-100 models take much less time, and on average, one can train each model in less than an hour.

### C.4   Results on CIFAR-100 Dataset

For the CIFAR-100 dataset, we computed the generalization gap ( by subtracting test loss on augmented CIFAR100-C data from train loss on the original CIFAR-100 dataset) for ResNet-18, MobileNet-V1, and VGG-19. Unlike the Imagenet-C dataset, the CIFAR100-C dataset comes with only one distortion severity level. Table 5 shows the result of our experiment on the CIFAR-100 dataset. Compared to full-precision models, quantized models show smaller generalization gaps in all cases.

### C.5   Results on ImageNet Dataset

For the ImageNet dataset, we computed the generalization gap ( by subtracting test loss on augmented data from train loss on the original ImageNet dataset) for ResNet-18, MobileNet V2, and ResNet-50. Table 6 shows the full list of experiments. As seen, compared to the full precision model and unlike the CIFAR-100 dataset, not all quantization levels show a better generalization gap. Especially for the MobileNet-V2 model, Int2 quantization shows the worst generalization gap in most distortion types and severity levels. But in general, Int8 and Int4 show better generalization gaps in almost all models, distortion types, and levels.

Table 5: Full list of experiments showing the effect of distortion on generalization gap on quantized models on CIFAR-100 dataset. Compared to FP32 column, we have highlighted better generalization gap with ▇ and ▇ to show the opposite.

| | ResNet-18 | | | | MobileNet-V1 | | | | VGG-19 | | | |
|---|---|---|---|---|---|---|---|---|---|---|---|---|
| Augmentation | FP32 | Int8 | Int4 | Int2 | FP32 | Int8 | Int4 | Int2 | FP32 | Int8 | Int4 | Int2 |
| Gaussian Noise | 2.782 | 1.546 | 1.614 | 1.225 | 4.327 | 0.022 | 0.027 | 0.191 | 4.26 | 1.687 | 0.91 | 0.497 |
| Shot Noise | 2.027 | 1.336 | 1.4 | 1.041 | 3.357 | 0.023 | 0.773 | 0.147 | 3.267 | 1.299 | 0.701 | 0.427 |
| Impulse Noise | 2.004 | 1.155 | 1.252 | 1.004 | 2.921 | 0.023 | 0.633 | 0.156 | 4.205 | 1.369 | 0.964 | 0.563 |
| Defocus Noise | 1.004 | 0.752 | 0.815 | 0.598 | 1.889 | 0.023 | 0.479 | 0.057 | 1.89 | 0.501 | 0.401 | 0.267 |
| Glass Blue | 4.649 | 2.445 | 2.479 | 1.889 | 5.849 | 0.022 | 0.186 | 0.23 | 8.709 | 2.369 | 1.457 | 0.797 |
| Motion Blur | 1.373 | 0.923 | 0.971 | 0.706 | 2.386 | 0.025 | 1.001 | 0.067 | 2.413 | 0.687 | 0.54 | 0.335 |
| Zoom Blur | 1.49 | 0.915 | 0.979 | 0.736 | 2.657 | 0.022 | 0.224 | 0.068 | 2.579 | 0.788 | 0.581 | 0.351 |
| Snow | 1.409 | 0.769 | 0.853 | 0.666 | 2.433 | 0.022 | 0.26 | 0.085 | 2.653 | 0.709 | 0.445 | 0.256 |
| Frost | 1.473 | 0.487 | 0.581 | 0.4 | 2.465 | 0.023 | 0.245 | 0.003 | 2.506 | 0.439 | 0.115 | 0.005 |
| Fog | 0.991 | 0.505 | 0.583 | 0.409 | 1.858 | 0.022 | 0.056 | 0.007 | 1.88 | 0.338 | 0.238 | 0.145 |
| Brightness | 0.996 | 0.543 | 0.609 | 0.435 | 1.869 | 0.022 | 0.048 | 0.01 | 1.878 | 0.349 | 0.229 | 0.112 |
| Contrast | 1.018 | 0.532 | 0.618 | 0.434 | 1.905 | 0.022 | 0.081 | 0.015 | 1.907 | 0.365 | 0.269 | 0.179 |
| Elastic | 1.415 | 0.996 | 1.048 | 0.828 | 2.493 | 0.022 | 0.055 | 0.113 | 2.507 | 0.8 | 0.646 | 0.429 |
| Pixelate | 1.185 | 0.888 | 0.982 | 0.767 | 2.171 | 0.022 | 0.336 | 0.087 | 2.136 | 0.713 | 0.479 | 0.33 |
| JPEG | 1.7 | 1.217 | 1.313 | 1.002 | 2.75 | 0.022 | 0.309 | 0.119 | 2.868 | 1.12 | 0.703 | 0.427 |

Table 6: Full list of experiments showing the effect of distortion on generalization gap on quantized models on ImageNet dataset. Compared to FP32 column, we have highlighted better generalization gap with ▇ and ▇ to show the opposite.

| Model | Augmentation | Severity 1 | | | | Severity 2 | | | | Severity 3 | | | | Severity 4 | | | | Severity 5 | | | |
|---|---|---|---|---|---|---|---|---|---|---|---|---|---|---|---|---|---|---|---|---|---|
| | | FP32 | Int8 | Int4 | Int2 | FP32 | Int8 | Int4 | Int2 | FP32 | Int8 | Int4 | Int2 | FP32 | Int8 | Int4 | Int2 | FP32 | Int8 | Int4 | Int2 |
| ResNet-18 | Gaussian Noise | 1.067 | 0.86 | 0.939 | 1.21 | 1.913 | 1.46 | 1.629 | 2.201 | 3.439 | 2.52 | 2.796 | 3.658 | 5.529 | 3.96 | 4.352 | 5.115 | 7.653 | 5.62 | 6.117 | 6.093 |
| | Shot Noise | 1.238 | 0.96 | 1.054 | 1.333 | 2.289 | 1.72 | 1.887 | 2.421 | 3.801 | 2.75 | 2.987 | 3.685 | 6.438 | 4.4 | 4.739 | 5.304 | 7.919 | 5.41 | 5.805 | 6.003 |
| | Impulse Noise | 2.235 | 1.78 | 2.058 | 2.324 | 3.177 | 2.35 | 2.636 | 3.279 | 4.061 | 2.9 | 3.185 | 4.001 | 6.096 | 4.26 | 4.595 | 5.324 | 7.781 | 5.57 | 5.995 | 6.114 |
| | Defocus Noise | 0.979 | 0.89 | 0.858 | 0.822 | 1.432 | 1.37 | 1.325 | 1.302 | 2.394 | 2.34 | 2.263 | 2.217 | 3.285 | 3.19 | 3.099 | 2.934 | 3.983 | 3.92 | 3.808 | 3.498 |
| | Glass Blue | 1.18 | 1.07 | 1.031 | 0.985 | 1.969 | 1.86 | 1.812 | 1.804 | 3.822 | 3.83 | 3.727 | 3.54 | 4.221 | 4.25 | 4.15 | 3.909 | 4.652 | 4.62 | 4.542 | 4.139 |
| | Motion Blur | 0.687 | 0.55 | 0.542 | 0.509 | 1.37 | 1.22 | 1.245 | 1.261 | 2.521 | 2.42 | 2.454 | 2.381 | 3.7 | 3.64 | 3.678 | 3.412 | 4.285 | 4.23 | 4.275 | 3.875 |
| | Zoom Blur | 1.518 | 1.38 | 1.382 | 2.219 | 2.1 | 2.119 | 2.094 | 2.688 | 2.58 | 2.588 | 2.518 | 3.213 | 3.12 | 3.137 | 3.028 | 3.666 | 3.59 | 3.599 | 3.437 | |
| | Snow | 1.401 | 1.01 | 0.998 | 1.124 | 3.215 | 2.36 | 2.374 | 2.643 | 2.969 | 2.07 | 2.094 | 2.315 | 3.97 | 2.81 | 2.869 | 3.074 | 4.515 | 3.48 | 3.517 | 3.572 |
| | Frost | 0.949 | 0.66 | 0.626 | 0.633 | 2.093 | 1.68 | 1.681 | 1.812 | 2.978 | 2.53 | 2.553 | 2.688 | 3.141 | 2.74 | 2.766 | 2.893 | 3.713 | 3.31 | 3.362 | 3.447 |
| | Fog | 0.809 | 0.42 | 0.444 | 0.405 | 1.214 | 0.68 | 0.734 | 0.774 | 1.857 | 1.18 | 1.273 | 1.431 | 2.347 | 1.68 | 1.762 | 1.958 | 3.77 | 3.03 | 3.141 | 3.275 |
| | Brightness | 0.121 | 0.04 | 0.019 | 0.155 | 0.221 | 0.1 | 0.08 | 0.062 | 0.378 | 0.19 | 0.184 | 0.084 | 0.631 | 0.37 | 0.36 | 0.323 | 0.986 | 0.62 | 0.626 | 0.672 |
| | Contrast | 0.523 | 0.24 | 0.232 | 0.13 | 0.867 | 0.4 | 0.413 | 0.396 | 1.627 | 0.81 | 0.861 | 1.031 | 3.61 | 2.37 | 2.529 | 2.921 | 5.264 | 4.63 | 4.765 | 4.479 |
| | Elastic | 0.538 | 0.43 | 0.406 | 0.287 | 2.026 | 1.95 | 1.911 | 1.833 | 1.116 | 1.03 | 0.969 | 0.884 | 1.997 | 1.94 | 1.844 | 1.755 | 4.112 | 4.11 | 3.957 | 3.57 |
| | Pixelate | 0.612 | 0.5 | 0.492 | 0.416 | 0.599 | 0.51 | 0.506 | 0.465 | 1.889 | 1.72 | 1.734 | 1.958 | 3.046 | 2.93 | 2.88 | 3.306 | 3.369 | 3.32 | 3.313 | 3.51 |
| | JPEG | 0.59 | 0.48 | 0.468 | 0.375 | 0.801 | 0.68 | 0.674 | 0.627 | 0.972 | 0.85 | 0.841 | 0.824 | 1.599 | 1.46 | 1.446 | 1.491 | 2.615 | 2.43 | 2.405 | 2.487 |
| MobileNet V2 | Gaussian Noise | 1.041 | 0.76 | 0.857 | 2.78 | 1.923 | 1.382 | 1.536 | 3.755 | 3.425 | 2.5 | 2.762 | 5.009 | 5.251 | 4.065 | 4.518 | 6.182 | 6.997 | 5.815 | 6.423 | 7.124 |
| | Shot Noise | 1.132 | 0.843 | 1.027 | 2.926 | 2.214 | 1.591 | 1.846 | 3.975 | 3.67 | 2.624 | 3.013 | 5.058 | 5.891 | 4.363 | 5.011 | 6.36 | 7.045 | 5.418 | 6.137 | 6.96 |
| | Impulse Noise | 1.635 | 1.483 | 1.585 | 3.043 | 2.597 | 2.223 | 2.284 | 4.144 | 3.423 | 2.751 | 2.901 | 4.961 | 5.302 | 4.171 | 4.591 | 6.296 | 6.979 | 5.753 | 6.315 | 7.126 |
| | Defocus Noise | 0.863 | 0.799 | 1.005 | 3.856 | 1.326 | 1.266 | 1.519 | 4.286 | 2.23 | 2.213 | 2.434 | 4.858 | 3.059 | 2.983 | 3.328 | 5.119 | 3.784 | 3.655 | 4.2 | 5.303 |
| | Glass Blue | 1.223 | 1.141 | 1.509 | 3.538 | 2.115 | 2.039 | 2.443 | 4.257 | 4.01 | 4.003 | 4.309 | 4.943 | 4.375 | 4.353 | 4.564 | 5.039 | 4.668 | 4.601 | 4.709 | 5.141 |
| | Motion Blur | 0.643 | 0.53 | 0.641 | 3.068 | 1.335 | 1.209 | 1.354 | 3.768 | 2.392 | 2.282 | 2.435 | 4.349 | 3.5 | 3.418 | 3.588 | 4.774 | 4.108 | 4.028 | 4.235 | 4.983 |
| | Zoom Blur | 1.539 | 1.423 | 1.607 | 3.564 | 2.282 | 2.185 | 2.358 | 3.97 | 2.774 | 2.685 | 2.886 | 4.335 | 3.317 | 3.263 | 3.492 | 4.567 | 3.797 | 3.738 | 4.035 | 4.817 |
| | Snow | 1.253 | 0.904 | 1.168 | 2.377 | 3.074 | 2.429 | 2.694 | 4.017 | 2.838 | 2.16 | 2.497 | 3.869 | 3.775 | 2.945 | 3.286 | 4.797 | 4.562 | 3.776 | 3.996 | 5.151 |
| | Frost | 0.941 | 0.658 | 0.8 | 2.236 | 2.193 | 1.784 | 2.021 | 3.713 | 3.154 | 2.673 | 2.969 | 4.682 | 3.345 | 2.904 | 3.223 | 4.94 | 3.956 | 3.484 | 3.835 | 5.46 |
| | Fog | 0.699 | 0.354 | 0.822 | 3.874 | 1.084 | 0.624 | 1.24 | 4.454 | 1.715 | 1.145 | 1.802 | 4.929 | 2.256 | 1.675 | 2.111 | 4.969 | 3.792 | 3.116 | 3.298 | 5.371 |
| | Brightness | 0.034 | 0.05 | 0.019 | 1.342 | 0.143 | 0.008 | 0.089 | 1.359 | 0.595 | 0.117 | 0.211 | 1.549 | 0.631 | 0.303 | 0.422 | 1.987 | 1.002 | 0.591 | 0.752 | 2.663 |
| | Contrast | 0.482 | 0.188 | 0.816 | 3.717 | 0.846 | 0.394 | 1.513 | 4.571 | 1.624 | 0.899 | 3.095 | 5.556 | 3.66 | 2.725 | 5.816 | 6.396 | 5.411 | 4.881 | 6.505 | 6.59 |
| | Elastic | 0.442 | 0.347 | 0.481 | 2.396 | 1.973 | 1.871 | 2.105 | 4.012 | 1.116 | 1.03 | 1.113 | 2.474 | 1.913 | 1.807 | 2.21 | 2.959 | 4.106 | 3.982 | 4.693 | 4.036 |
| | Pixelate | 0.926 | 0.653 | 0.872 | 1.883 | 1.444 | 1.02 | 0.934 | 1.838 | 2.155 | 1.822 | 2.468 | 2.172 | 3.111 | 3.064 | 3.773 | 2.755 | 3.979 | 3.993 | 3.988 | 3.301 |
| | JPEG | 0.491 | 0.382 | 0.554 | 1.754 | 0.675 | 0.552 | 0.784 | 1.848 | 0.826 | 0.693 | 0.967 | 1.928 | 1.357 | 1.165 | 1.555 | 2.182 | 2.182 | 1.902 | 2.462 | 2.545 |
| ResNet-50 | Gaussian Noise | 0.938 | 0.914 | 0.928 | 0.973 | 1.437 | 1.282 | 1.112 | 1.571 | 2.363 | 2.047 | 2.513 | 2.89 | 3.719 | 3.255 | 3.754 | 4.88 | 5.134 | 4.999 | 5.339 | 7.828 |
| | Shot Noise | 0.961 | 0.946 | 0.957 | 1.026 | 1.585 | 1.408 | 1.246 | 1.83 | 2.448 | 2.166 | 2.023 | 3.215 | 4.084 | 3.697 | 3.887 | 5.748 | 4.924 | 4.919 | 5.229 | 7.656 |
| | Impulse Noise | 1.703 | 1.652 | 1.676 | 1.789 | 2.013 | 1.874 | 1.464 | 2.373 | 2.564 | 2.295 | 1.995 | 3.211 | 3.962 | 3.507 | 3.458 | 5.435 | 5.28 | 4.942 | 5.105 | 8.123 |
| | Defocus Noise | 1.059 | 1.042 | 0.911 | 0.869 | 1.441 | 1.414 | 1.309 | 1.298 | 2.344 | 2.311 | 2.286 | 2.231 | 3.244 | 3.225 | 3.226 | 3.164 | 4.052 | 4.049 | 4.019 | 3.994 |
| | Glass Blue | 1.349 | 1.27 | 1.011 | 1.457 | 2.297 | 2.169 | 1.829 | 2.088 | 4.613 | 4.551 | 4.185 | 4.346 | 5.057 | 5.009 | 4.815 | 4.793 | 5.399 | 5.376 | 5.34 | 5.102 |
| | Motion Blur | 0.731 | 0.638 | 0.623 | 1.314 | 1.307 | 1.19 | 1.238 | 2.563 | 2.551 | 2.337 | 2.501 | 4.148 | 3.96 | 4.108 | 4.285 | 5.048 | 5.033 | 4.404 | 4.75 | |
| | Zoom Blur | 1.509 | 1.473 | 1.261 | 1.361 | 2.252 | 2.187 | 2.037 | 2.134 | 2.822 | 2.736 | 2.571 | 2.697 | 3.44 | 3.337 | 3.139 | 3.325 | 4.039 | 3.949 | 3.676 | 3.916 |
| | Snow | 1.229 | 1.13 | 1.048 | 1.143 | 2.62 | 2.529 | 2.481 | 2.933 | 2.375 | 2.317 | 2.193 | 2.478 | 3.127 | 3.016 | 2.941 | 3.347 | 3.697 | 3.437 | 3.8 | 4.209 |
| | Frost | 0.845 | 0.837 | 0.674 | 0.653 | 1.769 | 1.726 | 1.506 | 1.785 | 2.563 | 2.522 | 2.245 | 2.588 | 2.761 | 2.72 | 2.435 | 2.816 | 3.322 | 3.291 | 2.972 | 3.427 |
| | Fog | 0.691 | 0.685 | 0.582 | 0.501 | 0.897 | 0.876 | 0.784 | 0.926 | 1.305 | 1.248 | 1.167 | 1.337 | 1.81 | 1.657 | 1.635 | 1.909 | 3.261 | 2.96 | 2.979 | 3.569 |
| | Brightness | 0.345 | 0.25 | 0.21 | 0.081 | 0.382 | 0.314 | 0.259 | 0.431 | 0.453 | 0.446 | 0.341 | 0.222 | 0.584 | 0.514 | 0.473 | 0.367 | 0.787 | 0.716 | 0.674 | 0.609 |
| | Contrast | 0.545 | 0.449 | 0.42 | 0.39 | 0.69 | 0.677 | 0.568 | 0.494 | 1.047 | 0.998 | 0.867 | 1.051 | 2.387 | 2.173 | 1.85 | 2.624 | 4.881 | 4.394 | 4.686 | 4.914 |
| | Elastic | 0.655 | 0.625 | 0.517 | 0.422 | 2.274 | 2.243 | 1.908 | 2.122 | 1.602 | 1.571 | 1.242 | 1.416 | 2.704 | 2.671 | 2.209 | 2.591 | 5.598 | 5.522 | 4.647 | 5.348 |
| | Pixelate | 0.871 | 0.71 | 0.727 | 0.684 | 1.042 | 1.021 | 1.015 | 0.778 | 1.971 | 1.575 | 1.556 | 1.974 | 3.373 | 3.229 | 3.208 | 3.431 | 4.038 | 4.014 | 3.996 | 4.179 |
| | JPEG | 0.793 | 0.701 | 0.67 | 0.522 | 0.957 | 0.98 | 0.832 | 0.704 | 1.092 | 1.023 | 0.96 | 0.849 | 1.537 | 1.425 | 1.356 | 1.394 | 2.236 | 2.118 | 1.967 | 2.228 |

