# OpenReview forum: "QGen: On the Ability to Generalize in Quantization Aware Training"
_TMLR — Rejected by TMLR_

### Review · Reviewer_kpuY · 2024-05-11

**Summary Of Contributions:**

Summary: This paper argues that quantization can be seen as a regularizer, and provides empirical evidence supporting this claim, finding that quantization-aware training often (but not always) converges to parameters which generalize better than their full-precision counter-parts, lie in flatter minima, and are more robust to perturbations.

**Audience:**

Yes

**Claims And Evidence:**

No

**Requested Changes:**

Requested changes:

- The claims in Section 1 should be adjusted to reflect the contributions of the paper.

- The theoretical analysis in Section 3 should be revised to reflect the approximation error incurred by the first-order Taylor expansion and justify the noisy perturbation model.

 - The paper should provide some intuition for why quantization does not uniformly increase robustness/flat minima as the precision decreases.

- The paper should contrast quantization-aware training with an equivalent noise model which applies uniform random perturbations to the gradient updates to validate the proposed regularization mechanism.

**Strengths And Weaknesses:**

Strengths:

- As the authors point out in section 2, while the community has historically been in agreement that the noise introduced by quantization will likely benefit the generalization properties of the network, this effect has not been rigorously studied empirically and the paper is thus well-positioned to provide new insights into this phenomenon.

- The empirical evaluations are thorough and transparent, and data which disagree with the paper's hypothesis are not omitted.

- The paper is clearly written and easy to follow.

Weaknesses:

- The claims made in the introduction are not entirely supported by the paper's results.
  1. The claim that the paper 'theoretically shows' that quantization can be seen as a regularizer is not entirely accurate. Section 3.1 approximates the loss with a first-order Taylor approximation and then proceeds to write a series of equalities which are incorrect without accounting for higher-order terms in the expansion. Further, the derivation assumes that the quantization noise is independent (or possibly orthogonal, it's unclear what the \perp symbol is being used for) of the gradient without proof or reference to any justification for this assumption. Further, the resulting claim is that quantization is identical to applying uniform random perturbations with bin width $\delta$. This would be an easy corollary to test empirically, and it is surprising that uniform random perturbations are not included as a baseline.

  2. The claim to "empirically show that there exists a quantization level at which the quantized model converges..." is not technically wrong but should be made more precise. Specifically, the paper does not show that there exists a quantization such that for any model, the quantization-aware training process will converge to a flatter minimum. Instead, the paper observes that for the models studied, there is always at least one quantization level for which a particular model will reach flatter minima than its full-precision baseline, although this level varies between models.

  3. The paper shows that quantized models *usually* exhibit greater robustness to distorted data, but not always. Further, for any particular quantization level there will be some models where this quantization improves robustness and others where it harms robustness.


- Given the above nuances, it is difficult to determine what conclusions to draw from the paper. If quantization is equivalent to adding random noise to the optimization process, then why is this effect so non-uniform, with greater levels of quantization often leading to *sharper* minima? Why do we see similar non-uniform effects with respect to input distortions and the generalization gap? I suspect that part of the answer to this is that quantization-aware training introduces additional factors into the training process which are glossed over in the theoretical analysis.

---

> ### Author Response · Authors · 2024-07-01
>
> We would like to thank kpuY for their detailed comments and feedback. We are glad that they have found our work “well-positioned to provide new insights into this phenomenon.”
>
> 1. The claims in Section 1 should be adjusted to reflect the contributions of the paper.
>
> We have adjusted our claims as you suggested to “Our theoretical analysis on simplified models suggests that quantization can be seen as a regularizer”. This modification aligns with your feedback while maintaining the integrity and overall outcome of the paper.
> Regarding Taylor approximation:
> It is common practice in machine learning and mathematics to use first order Taylor approximation. By definition, Taylor approximation is not exact and one can disregard higher order terms. This does not mean that the equalities are incorrect, but rather, as mentioned in our work, it is an approximation. We agree that this approximation is not exact. We will emphasize this in the manuscript.
>
> 2. Regarding the noise model:
>
> We would like to remind the reviewer that we based our noise model on previous work (as already cited in our paper). See [1, 2, 3].
> Furthermore, as mentioned in the footnote on page 4, the only requirement that we have on the noise is to have zero-mean. As a result, the Uniform assumption could be relaxed but we followed the literature in this regard.
>
> 3. The paper should provide some intuition for why quantization does not uniformly increase robustness/flat minima as the precision decreases.
>
> In this work, we examine the relationship between quantization and regularization, showing that quantization can be seen as a regularizer that requires tuning. We argue that quantization hyperparameter (precision) should be treated as a critical component of model training, alongside traditional regularizers, as each model and dataset presents a unique optimization challenge. Our results indicate that the relationship between quantization precision and regularization is more complex than previously thought, and that uniform reductions in precision do not necessarily lead to improved robustness or flat minima. Hence, we do not expect that reducing the quantization precision uniformly correlates with better robustness/flat minima.
>
> The paper should contrast quantization-aware training with an equivalent noise model which applies uniform random perturbations to the gradient updates to validate the proposed regularization mechanism.
>
> 4. In [4], it has been demonstrated that zero-mean noise acts as a regularization technique that enhances model generalization. As noted in our earlier remarks on page 4, footnote, this aligns precisely with the type of noise we are considering. Given the established nature of this phenomenon, we deem further exploration unnecessary.
>
> [1] Alexandre Défossez, Yossi Adi, and Gabriel Synnaeve. Dierentiable model compression via pseudo quantization noise. arXiv preprint arXiv:2104.09987, 2021.
>
> [2] Bernard Widrow, Istvan Kollar, and Ming-Chang Liu. Statistical theory of quantization. IEEE Transactions on instrumentation and measurement, 45(2):353–361, 1996.
>
> [3] Eirikur Agustsson and Lucas Theis. Universally quantized neural compression. Advances in neural information processing systems, 33:12367–12376, 2020
>
> [4] Graves, Alex. “Practical variational inference for neural networks.” Advances in neural information processing systems. Vol. 24. 2011.

---

### Review · Reviewer_QvcK · 2024-06-05

**Summary Of Contributions:**

This paper focuses on neural network quantisation, and specifically to what extent quantised neural network generalise well between training and testing or with noised-up data. The main conclusion that the paper draws is that quantization not only helps with making neural networks more efficient, but also allow for flutter minima, thus improving generalization to a good extent.

**Audience:**

Yes

**Broader Impact Concerns:**

Not relevant.

**Claims And Evidence:**

Yes

**Requested Changes:**

Please see the weaknesses above and comment accordingly.

**Strengths And Weaknesses:**

The paper has a solid technical analysis with bounds on the effect of quantization and quantized weights to the final loss. Specifically, under a probabilistic framework quantization is viewed as a stochastic process of adding to the on network weights uniform noise. the question itself of studying the effect of quantization generalization is of course of important one, especially for practitioners.

The derived bounds make it easy to understand the free variables that contribute to better or worse and generalization given quantization. Specifically, the bounds link the magnitude and the range of noise to the final training loss. What is a particular relevance is the fact that the paper stresses that the magnitude of the weights is just as important when examining the generalization of the loss landscape, and specifically the sharpness or flatness of it.

Another strong point is the extensive experimental validation on three data, including image Imagenet. What is more, the paper includes ablation studies with respect to perturbations to the data. This shows that quantized networks do not only generalize well with respect to training and testing distributions, but also with respect to noised up data.

Regarding weaknesses, I cannot think of many since this looks like a solid work. However, I would say that I would like a bit more clarity on how the PAC-Bayesian and Sharpness measures are explicitly computed. I know there are references included, however, it would be more comfortable if this is directly explained in the paper, considering also the relevance for interpreting the results.

What is more, I would welcome a bit more clarity on notation. For instance, L_S is the loss in the training set, from what I understand, and then L_D is the loss if one had access to the underlying theoretical data distribution? Where is the loss in the test set, according to which the generalization gaps are reported?

Perhaps more importantly, I wonder how loose or tight the bounds are? Since \rho is a free variable in 12-14, a small rho would yield not very flexible quantization but a tighter bound, however, more flexible quantization (eg with more bits) would give looser bounds? Do I interpret this correctly? Does this make sense?

---

> ### Author Response · Authors · 2024-07-01
>
> We would like to thank QvcK for their detailed comments and feedback. We are glad that they have found our work “The paper has a solid technical analysis with bounds on the effect of quantization and quantized weights to the final loss.”
>
> 1. I would say that I would like a bit more clarity on how the PAC-Bayesian and Sharpness measures are explicitly computed. I know there are references included, however, it would be more comfortable if this is directly explained in the paper, considering also the relevance for interpreting the results.
>
> We have added an appendix to our paper that details the methodology used to calculate these metrics in our experiments. Please see Appendix A. This appendix was initially included as part of the “Supplementary Material” in our paper submission, but for convenience, it has now been integrated into the main paper.
> Additionally, we have created a GitHub repository containing the source code for all our experiments, which offers a practical demonstration of how we computed sharpness in our research. The source code is also provided as a zip file in the “Supplementary Material.”
>
> 2. I would welcome a bit more clarity on notation. For instance, L_S is the loss in the training set, from what I understand, and then L_D is the loss if one had access to the underlying theoretical data distribution? Where is the loss in the test set, according to which the generalization gaps are reported?
>
> Yes L_S is the loss in the training set and L_D is the loss of the model in the data space. In the context of our discussion in section 3, our aim was to demonstrate that the quantized model adds a regularization term to the loss function during training, which is why we did not discuss the test set. In section 4, we talked about what proxy measures we used for generalization. It is only in experiments that we talked about actual generalization with respect to the test set. In Table 1, where we provide generalization gap measurements, we have provided both test loss and training loss.
>
> 3. Perhaps more importantly, I wonder how loose or tight the bounds are? Since \rho is a free variable in 12-14, a small rho would yield not very flexible quantization but a tighter bound, however, more flexible quantization (e.g. with more bits) would give looser bounds? Do I interpret this correctly? Does this make sense?
>
> In  [1] the authors highlight the inherent difficulty in establishing tight measures for generalization. While we cannot definitively confirm the tightness of these bounds, we substantiate our claims through rigorous experiments detailed in Section 4. These experiments provide empirical support for our assertions, even though the challenge of achieving optimal bounds remains.
>
> [1] Jiang, Y., Neyshabur, B., Mobahi, H., Krishnan, D., & Bengio, S. (2019). Fantastic generalization measures and where to find them. arXiv preprint arXiv:1912.02178.

---

### Review · Reviewer_miLU · 2024-06-17

**Summary Of Contributions:**

This work performs an empirical study of the effect of network quantization on generalization. Specifically, it looks at the generalization gap for bitwidths of 2, 4, 8 bits compared to floating point performance, and finds that
- the generalization gap is typically smaller
- the loss landscape is more flat than for floating point networks

The authors argue that quantization can be seen as a form of regularization, rewriting the loss as a default (float) loss and an additional term depending on the quantization noise. This allows connecting the "quantization rate" or in this case bitwidth, to the regularization level of quantization.

They also show that under a number of conditions, it is possible to make some claims about the flatness of the minima of the loss, but there are no real guarantees, if I understand their hypothesis 1 correctly.

Another main contribution is the evaluation of a large number of models under various amounts of quantization, and demonstrating that the generalization gap and difference in flatness exist in practice.

**Audience:**

Yes

**Broader Impact Concerns:**

N/A, although quantization can increase biases in networks this work is not specifically investigating those.

**Claims And Evidence:**

Yes

**Requested Changes:**

Please consider adding
- a more thorough explanation of the loss landscape figures of Figure 2. What can we see, and why is the landscape flatter as we increase quantization level? Naively, 2-it looks like it has a quite sharp minimum too?
- in section 4.1, it is mentioned that flatter loss landscapes are an indicator of better generalization. Can you add some intuition or evidence for why this is the case?

**Strengths And Weaknesses:**

The main strength of this work is the huge number of evaluated models support the authors claims, but also an - as far as I can tell correct - potential explanation for why we see flatter minima when using quantization, even it comes with many assumptions and some caveats.
- an important observation: that measuring flatness along different quantization levels would benefit from taking the weight intensities into account

Some weaknesses include
- theoretical analysis performed only for scale-only quantization with equal bin size
- observations aren't always consistent, especially for the augmentation results

Of course, the observation that the generalization gap and loss landscape change as different quantization levels are applied does not make it into a practical algorithm that can actually improve performance. Nevertheless, I think this study adds to the empirical and theoretical studies of quantization.

---

> ### Author Response · Authors · 2024-07-01
>
> We would like to thank miLU for their detailed comments and feedback. We are glad that they have found our work valuable, especially noting that “The main strength of this work is the huge number of evaluated models support the authors claims.”
>
> 1. A more thorough explanation of the loss landscape figures of Figure 2. What can we see, and why is the landscape flatter as we increase quantization level? Naively, 2-bit looks like it has a quite sharp minimum too?
>
> The sharpness measurements detailed in Table 3 indicate that quantization aware training leads to flatter minima. To visually represent this, we examined the loss landscape of quantized models and a full precision model around the minima, as depicted in Figure 2. These minima were obtained after complete training of the models. Following training, we utilized the trained model and a method described in detail by [1] to explore the vicinity of the minima. We then projected the loss from a high-dimensional space into a 3-dimensional graph. The results show that compared to full precision models, the 8 and 4 bit quantized models exhibit flatter minima. However, as noted, the 2-bit model displays a sharp minima, which aligns with our findings as presented in the paper. We posit that quantization aware training serves as a form of regularization and, like any regularization technique, must be finely tuned to achieve optimal generalization.
>
> 2. In section 4.1, it is mentioned that flatter loss landscapes are an indicator of better generalization. Can you add some intuition or evidence for why this is the case?
>
> The relationship between loss landscape and generalization is still an active area of research, the idea that flatter loss landscapes are associated with better generalization is supported by both theoretical intuitions and empirical evidence. The most principled way to measure generalization is to find a generalization bound but with the ever growing complexity of neural networks, finding tight bounds has been shown to be very difficult. However, it is important to note that sharpness is a proxy measure for generalization. As pointed out in [2], there are other such proxies to measure generalization and among them sharpness measures have shown good correlation with generalization. Apart from the empirical studies that have shown the correlation of better generalization and sharpness [2, 3], there are intuitions on this phenomenon as well. In [4] authors show that Flatter minima are associated with better generalization because they occupy larger volumes in high-dimensional parameter spaces. Intuitively, during training, stochastic optimizers like SGD are more likely to land in these wide basins. In [5], discuss how flatter minima are less sensitive to small perturbations in the input data or model parameters which implies that the model's performance does not degrade significantly with slight changes, which is a desirable property for generalization.
>
>
> [1] Li, H., Xu, Z., Taylor, G., Studer, C., & Goldstein, T. (2018). Visualizing the loss landscape of neural nets. In Advances in Neural Information Processing Systems, Vol. 31,  2018.
>
> [2] Jiang, Y., Neyshabur, B., Mobahi, H., Krishnan, D., & Bengio, S. (2019). Fantastic generalization measures and where to find them. arXiv preprint arXiv:1912.02178.
>
> [3] Foret, P., Kleiner, A., Mobahi, H., & Neyshabur, B. (2021). Sharpness-aware minimization for efficiently improving generalization. In the International Conference on Learning Representations.
>
> [4] Zhu, C., Huang, W. R., Li, H., Taylor, G., Studer, C., & Goldstein, T. (2019). Transferable clean-label poisoning attacks on deep neural nets. In K. Chaudhuri & R. Salakhutdinov (Eds.), Proceedings of the 36th International Conference on Machine Learning (Vol. 97, pp. 7614-7623). PMLR
>
> [5] Zhang, J., Qi, L., Shi, Y., & Gao, Y. (2023). Exploring flat minima for domain generalization with large learning rates. ArXiv.

---

> > ### Comment · Reviewer_miLU · 2024-07-02
> > **Thanks for the response**
> >
> > 1. Clear, the figure is only intended to show that the loss for quantized ResNet18s looks flatter than full precision? I agree with reviewer kpuY that the claims made this observation seem more broad than it is. We see that for a ResNet18, different quantization levels result in flatter minima than a full precision loss landscape. But the subtext has a more general statement "quantized models possess flatter minima, which contributes to enhanced generalization capabilities". Especially that last part, on generalization capabilities, is not what the figure shows and is better left to the main text in my opinion.
> >
> > 2. Yes, the intro and section 2.2 reference other efforts to correlate sharpness and generalization (including [2, 3]). I was wondering if there were hypotheses for why this is the case, and whether these could be related to quantization. Thanks for reference [5], useful for intuition, maybe worth adding in 4.1?
> >
> > Reference [4] does conjecture that their attack is more effective because the volume of the polytope in which the target's feature lies is larger (and that it therefore tolerates larger generalization error), but I don't think they show that flatter minima are associated with better generalization? Correct me if I'm wrong here.
> >
> >
> > Lastly, I agree with reviewers m3NZ and kpuY that you should be careful of broad claims based on theoretical analysis if that analysis has caveats (quadratic loss only, approximate bounds, scale-only quantization with equal binwidth).

---

> > > ### Author Response · Authors · 2024-07-12
> > >
> > > 1. Thank you very much for your feedback. We agree that the caption for Figure 2 could be improved and we have revised it to better reflect the observations and claims presented in the paper and Figure 2.
> > >
> > > 2. To the best of our knowledge, there is no existing work that attempts to connect model flatness and quantization. We would be happy to review and compare any relevant research if you can provide references. Additionally, we have incorporated reference 5 into our paper. If there are other suggestions or related studies we should consider, please let us know, and we will be glad to address them.
> > >
> > > 3. Yes, we agree. A much better reference for showing association of flatter minima and generalization is [1] where the authors show the association of low complexity regions (flatter minima) and generalization.
> > >
> > > 4. Thank you for your feedback. We acknowledge the importance of careful claims when based on theoretical analysis with inherent limitations, such as quadratic loss, approximate bounds, and scale-only quantization with equal bin width. As discussed in previous replies, our use of first-order Taylor approximation is a common practice and its limitations were noted, we have thus emphasized these caveats in the manuscript to ensure clarity and avoid broad claims. We also want to highlight that our contributions have been adjusted to accurately reflect our underlying assumptions and simplifications.  Please refer to Section 1.
> > >
> > > [1] Wu, L., Zhu, Z., & Weinan, E. (2017). Towards Understanding Generalization of Deep Learning:      Perspective of Loss Landscapes. ArXiv, abs/1706.10239.

---

### Review · Reviewer_m3NZ · 2024-06-26

**Summary Of Contributions:**

This paper studies the relationship between generalization and quantization. More precisely, the paper examines Quantization Aware Training (QAT). It applies the per-layer quantization approach, where each target quantization layer learns a step size to quantize the layer weights.

The main observation of this paper involves examining a regression problem and showing that the quantization noise induces a gradient norm regularization term.

Next, this paper reviews the notion of sharpness from the work of Foret et al. (2021), and then it carries out a large-scale experiment for different quantization levels over 3 image datasets.

Lastly, the paper provides empirical evidence that on distorted data, quantized models exhibit improved generalization compared to their full-precision counterparts across various experimental setups.

**Audience:**

Yes

**Claims And Evidence:**

No

**Requested Changes:**

- For the claims of this paper to be more generalizable to a broader audience, it is suggested to run the experiments on different types of quantization methods. See below for some examples:

Zhao, R., Hu, Y., Dotzel, J., De Sa, C., & Zhang, Z. (2019, May). Improving neural network quantization without retraining using outlier channel splitting. In International conference on machine learning (pp. 7543-7552). PMLR.

De Sa, C., Feldman, M., Ré, C., & Olukotun, K. (2017, June). Understanding and optimizing asynchronous low-precision stochastic gradient descent. In Proceedings of the 44th annual international symposium on computer architecture (pp. 561-574).

- This paper only includes experiments on the CIFAR datasets. For the claims to be more relevant to a broader TMLR audience, it would be helpful to run the experiments on a broader range of datasets and applications.

- For the materials of Section 4, since much of it comes from Foret et al. (2021), it is suggested that the authors move the materials from Foret et al. (2021) to appendix or supplementary materials, and leave the original contributions in this section only.

- An extension of the theoretical analysis to other types of losses would better substantiate the claims of this paper. Besides, clarifying why having a penalty on the gradient norm would be considered a regularization as opposed to some form of implicit regularization directly from running SGD.

**Strengths And Weaknesses:**

Strengths:
- This paper studies a very relevant problem. As models are getting bigger, gaining a better understanding of quantization aware training is certainly a useful pursuit.

Weaknesses:
- My main concern is that the claims made in this paper are not clearly backed up by concrete evidences--this is based on TMLR's evaluation criterion.

For instance, the authors claim that We theoretically show that quantization can be seen as a regularizer. However, what they showed is that the quantization noise induces a regularization on the gradient norm.
- First, I am not sure if this should be called a "regularization." If SGD keeps running, it will eventually shrink the magnitude of the gradient.
- Second, the analysis is only carried out on a quadratic loss. What about cross-entropy loss?

Second, the authors claim that there exists a quantization level at which the quantized model converges to a flatter minimum than its full-precision model. However, this claim is missing statements regarding the test loss value. If I set the model at zero, then it is also very flat, but the training loss is pretty bad. At least, this statement should be clarified so that it adjusts for both test performance and flatness.

Third, the authors claim regarding distorted data requires setting up some context as this came out before it was described or motivated.

---

> ### Author Response · Authors · 2024-07-12
>
> We would like to thank m3NZ for their detailed comments and feedback. We are glad that they have found our work valuable, especially noting that “This paper studies a very relevant problem”. Please find our response to the requested changes below:
>
>
>
> 1. Thank you for your suggestions. As noted in the manuscript, we utilized two different quantization methods: LSQ and VVTQ. Although VVTQ is similar to LSQ, it is specifically tuned for quantizing transformer-based models, as explained in the paper. We believe that exploring the generalization of various quantization methods is an interesting and valuable study, but it falls outside the scope of this paper. Our focus has been on understanding generalization within Quantization-Aware Training (QAT), rather than determining which QAT method has better generalization. Although this is beyond the current scope of our paper, we intend to investigate it in future research.
>
>
> 2. We believe the reviewer might have overlooked some aspects of our results. We encourage a detailed examination of Tables 1, 2, 3, and the appendix. Our results are based on the CIFAR-10, CIFAR-100, and ImageNet datasets. We believe our extensive experiments across these diverse datasets and models provide comprehensive coverage of a wide range of applications.
>
>
> 3. We never claimed that the equations in section 4 are our original work and we have properly cited the reference papers. We believe to conduct a meaningful understanding of our claims in section 4, the reader requires some context and that is why we have added those equations (with reference to original work) in our manuscript.
>
>
> 4. Thank you for your feedback. We agree that extending the theoretical analysis to other types of losses would enhance the robustness of our claims. While this is beyond the current scope of our paper, we plan to explore this in future work.
>
> Regarding the implicit regularization of SGD vs the added penalty to the loss function through gradient norm. Explicitly adding a penalty on the gradient norm is a form of regularization because it directly modifies the objective function to encourage certain desirable properties in the learned model. Specifically, by penalizing the gradient norm, the model is encouraged to find flatter minima, which have been associated with better generalization performance. This is supported by several studies, such as [1].  On the other hand, the implicit regularization effects of stochastic gradient descent (SGD) stem from the noise introduced by the stochastic nature of the optimization process. Which may exhibit a form of implicit regularization that is separate from gradient norm regularization.
>
> Regarding the test loss, we have assumed that these analyses are conducted on well-trained models. Our claim about the quantized model converging to a flatter minimum than the full-precision model is made under the assumption that the models in question have been trained to a satisfactory level of performance. We recognize that flatness alone does not guarantee good generalization and, as mentioned in the paper, serves as a proxy measure for generalization. Therefore, our analysis ( please refer to Table 1 ) considers both the test performance and the flatness of the minima.
>
> [1] X. Zhang, R. Xu, H. Yu, H. Zou, and P. Cui, "Gradient Norm Aware Minimization Seeks First-Order Flatness and Improves Generalization," in *Proceedings of the IEEE/CVF Conference on Computer Vision and Pattern Recognition (CVPR)*, June 2023, pp. 20247-20257

---

> ### Comment · Reviewer_m3NZ · 2024-07-27
> **Response to Official Comments**
>
> Thanks for your response. I've read the reference that you suggested:
>
> - In Lemma 4.1 of Zhang et al. (2023), what they define is the *maximum* of the norm of the gradient, within a small vicinity of the neighborhood. But that refers to the worst-case direction that increases the gradient. But this is different from Equation (7), which is taken as the average over $p$.
>
> I've looked at Tables 1-3. What I'm asking is how this flatness affects training loss. Please note that this is different from measuring the generalization gap. Showing that the generalization gap is small *does not imply* that the method will automatically perform better. One has to take into account the effect of the training loss.
>
> In addition, I do not agree that extending the theoretical analysis to other types of losses is *beyond the current scope of our paper.* If the theoretical claim regarding the regularization of quantization only holds on quadratic losses, then experiments in this same setting would be expected. Otherwise there remains a gap in the arguments.

---

### Decision · Action_Editor_HL75 · 2024-07-25

**Recommendation:** Reject

**Comment:**

This paper investigates the generalization properties of quantized neural networks, claiming that lower-precision models generalize better. Specifically, the paper develops a theoretical model linking quantization as a form of regularization. The authors also conducted extensive experiments, empirically demonstrating that quantized models can converges to a flatter minimum than its full-precision counterparts.

The paper received mixed reviews, with recommendations of two accept and two reject. Overall, reviewers acknowledge several strengths of the paper:

1). Explores an important topic.

2). Paper is reasonably well-written and easy to follow.

2). The approach is interesting and valid. Particularly, generation bounds link the magnitude of the weights and the quantization noise to the loss.

3). The empirical evaluation is extensive.

However, the reviewer raised main concerns about the lack of clarity in the theoretical model and the inconclusive results. Here are a few examples:

1). The theoretical model connecting the quantization to a regularization term is only performed on quadratic losses and only assumes uniform quantization. It is unclear if this model can be extended to more general quantization aware training (QAT) cases.

2). The resulting regularization term is a non-typical, implicit gradient norm regularization. Since the paper claims to explain the generalization ability of QAT, it is expected that the authors provide more details on how this regularization behavior impacts on generalization.

3). The hypothesis is currently only supported by empirical evidence. Given the widespread use of QAT, a more variety of models and benchmarks are necessary to validate the conclusion. In addition, the observations are not always consistent and there is a lack of explanation.

4). Fail to provide intuition on the bound, such as whether it is tight or loose, parameter assumption, and why quantization does not uniformly increase robustness or result in flatter minima as the precision decreases.

During rebuttal, the authors did not provide further clarification and failed to address some reviewers’ main concerns.

In summary, while the paper explores an important topic and provides extensive experimental results, the lack of clarity in the theory and the inconclusiveness of the results make it difficult to evaluate its real impact. Therefore, I recommend borderline rejection.

**Audience:**

Yes. Understanding the generalization ability of quantized models is an important and relevant topic.

**Claims And Evidence:**

Not quite. Both theoretical evidence and experimental results are not clear enough to support the claims.

**Resubmission Of Major Revision:**

The authors may consider submitting a major revision at a later time.